

# Thetis coastal ocean model: discontinuous Galerkin discretization for the three-dimensional hydrostatic equations

Tuomas Kärnä[1], Stephan C. Kramer[3], Lawrence Mitchell[2,4], David A. Ham[2], Matthew D. Piggott[3], and António M. Baptista[1]

[1]NSF Science and Technology Center for Coastal Margin Observation & Prediction, Oregon Health & Science University, Portland, Oregon, USA
[2]Department of Mathematics, Imperial College London, London, United Kingdom
[3]Department of Earth Science and Engineering, Imperial College London, London, United Kingdom
[4]Department of Computing, Imperial College London, London, United Kingdom

*Correspondence to:* tuomas.karna@gmail.com

**Abstract.** Unstructured grid ocean models are advantageous for simulating the coastal ocean and river-estuary-plume systems. However, unstructured grid models tend to be diffusive and/or computationally expensive which limits their applicability to real life problems. In this paper, we describe a novel discontinuous Galerkin (DG) finite element discretization for the hydrostatic equations. The formulation is fully conservative and second-order accurate in space and time. Monotonicity of the advection scheme is ensured by using a strong stability preserving time integration method and slope limiters. Compared to previous DG models advantages include a more accurate mode splitting method, revised viscosity formulation, and new second-order time integration scheme. We demonstrate that the model is capable of simulating baroclinic flows in the eddying regime with a suite of test cases. Numerical dissipation is well-controlled, being comparable or lower than in existing state-of-the-art structured grid models.

## 1 Introduction

Numerical modeling of the coastal ocean is important for many environmental and industrial applications. Typical scenarios include modeling circulation at regional scales, coupled river-estuary-plume systems, river networks, lagoons, and harbors. Length scales range from some tens of meters in rivers and embayments to tens of kilometers in the coastal ocean; water depth ranges from less than a meter to kilometer scale at the shelf break. The time scales of the relevant processes range from minutes to hours, yet typical simulations span weeks or even decades. The dynamics are highly non-linear, characterized by local small-scale features such as fronts and density gradients, internal waves, and baroclinic eddies. These physical characteristics imply that coastal ocean modeling is intrinsically multi-scale, which imposes several technical challenges.

Most coastal ocean models solve the hydrostatic Navier-Stokes equations under the Boussinesq approximation – a valid approximation for mesoscale and sub-mesoscale (1 km) processes. Small-scale processes (< 100 m) are, however, inherently three-dimensional where non-hydrostatic effects can be important, especially in areas with pronounced density structure and stratification (Marshall et al., 1997b; Mahadevan, 2006). Non-hydrostatic modeling requires very high horizontal mesh resolu-



tion, which is currently only feasible in relatively small sub-regions (e.g. at the mouth of an estuary; Shi et al. 2016) due to its high computational cost.

Historically, regional ocean models have used structured, (deformed) rectilinear lattice grids. Although structured grids offer better computational performance (Danilov et al., 2008; Danilov, 2013), unstructured grids are generally preferred in coastal
domains as they can better represent the complex coastal topography and local features (Deleersnijder and Lermusiaux, 2008; Danilov, 2013; Piggott et al., 2013). Due to the large geometrical aspect ratio of the oceans (length versus depth), most models utilize computational grids that are layered in the vertical direction. Typical approaches include the terrain-following sigma levels (Blumberg and Mellor, 1987), equipotential $z$ levels (Griffies et al., 2005), isopycnal coordinates (Bleck, 1978), and their generalizations (e.g. Song and Haidvogel, 1994; Bleck, 2002).

In this article, we focus on solving the hydrostatic equations on an unstructured grid. While many unstructured grid models exist, their drawbacks tend to be excessive numerical diffusion that smooths out important physical features (Kärnä et al., 2015; Kärnä and Baptista, 2016; Ralston et al., 2017), and/or high computational cost. To address these issues, we propose a novel finite element solver for the hydrostatic equations, based on discontinuous Galerkin discretization methods.

Maintaining high numerical accuracy is crucial in ocean applications. The ocean is a forced dissipative system where the
mixing of water masses only takes place at the molecular level (Griffies, 2004). In practice, however, the finite grid resolution and numerical schemes used by the model introduce mixing rates of tracers and momentum that can be orders of magnitude larger than physical mixing (Burchard and Rennau, 2008; Rennau and Burchard, 2009; Hiester et al., 2014). Such spurious, numerical mixing is often dominated by the discretization of advection (Marchesiello et al., 2009; Griffies et al., 2000), but it can arise from other components as well, such as (implicit) time integration methods (Shchepetkin and McWilliams, 2005), or
various filters introduced to improve numerical stability (Danilov, 2012; Zhang et al., 2016).

In global circulation models, numerical mixing is a major bottleneck as (diapycnal) diffusion is very low in the deep ocean basins and water masses can remain largely unchanged for hundreds of years (Griffies, 2004; Griffies et al., 2000). Numerical mixing can, however, be a major issue in coastal domains as well: coastal oceans are characterized by strong density gradients, fronts between water masses (e.g. in river plumes), small-scale dynamics (e.g. internal waves and hydraulic jumps), and baro-
clinic eddies. An overly diffusive model can, therefore, fail to capture many essential physical features of these domains: it can smear out fronts, underestimate the intrusion of saline waters into embayments (Burchard and Rennau, 2008; Hofmeister et al., 2010; Kärnä et al., 2015; Ralston et al., 2017), or misrepresent mixing in river plumes.

The most common spatial discretization scheme is the finite volume (FV) method, used in MITgcm (Marshall et al., 1997a), GETM (Burchard and Bolding, 2002), ROMS (Shchepetkin and McWilliams, 2003, 2005), MPAS-Ocean (Ringler et al., 2013;
Petersen et al., 2015), UnTRIM (Casulli and Walters, 2000), FVCOM (Chen et al., 2003), SUNTANS (Fringer et al., 2006), FESOM2 (Danilov et al., 2016), and others. The FV method is well suited for advection-dominated problems, provides strict conservation of volume and mass, and yields good computational performance. FV methods are nominally only first-order accurate, but higher-order approximations can be introduced by increasing the size of the numerical stencil (e.g. in high-order advection schemes, Shchepetkin and McWilliams 1998).





Some unstructured grid models are based on the continuous Galerkin Finite Element (FE) method or hybrid FE-FV formulations. Such models include ADCIRC (Luettich and Westerink, 2004), SELFE (Zhang and Baptista, 2008), and SCHISM (Zhang et al., 2016), and the earlier version of FESOM (Wang et al., 2014). The continuous FE method is ideal for solving elliptic equations but requires stabilization for advection (see Wang et al., 2008a, and references therein). In addition, these

methods involve solving a fully-coupled global system which is less efficient in parallel applications compared to the FV method (Danilov, 2012; Danilov et al., 2016).

In recent years, discontinuous Galerkin (DG) methods have gained attention in geophysical modeling (Dawson and Aizinger, 2005; Aizinger and Dawson, 2007; Blaise et al., 2010; Comblen et al., 2010a; Kärnä et al., 2012, 2013). DG discretization resembles the FV method because it is local (i.e. elements are only connected by inter-element fluxes), fully conservative,

and well-suited for advective problems, yet it offers higher-order accuracy. This article presents a DG discretization for the hydrostatic equations. Our goal is to design an efficient unstructured grid solver where numerical accuracy is not compromised. Specifically, we aim to meet the following design criteria:

- a vertically extruded, layered mesh;

- accurate representation of free surface dynamics;

- second-order accurate, monotone tracer advection scheme;

- explicit time integration of 3D variables (except for vertical diffusion);

- and low numerical mixing.

Based on the advection scheme requirements, we have chosen to use linear discontinuous Galerkin elements for tracers, combined with a slope limiter (Kuzmin, 2010) and a strong stability preserving (SSP) time integration scheme (Shu, 1988;

Shu and Osher, 1988; Gottlieb and Shu, 1998; Gottlieb, 2005; Gottlieb et al., 2009). This choice ensures that the scheme is second-order in smooth areas, while slope limiting combined with the SSP time integration scheme ensure monotonicity (i.e. no overshoots). The movement of the free surface is taken into account with an arbitrary Lagrangian-Eulerian (ALE) formulation (Donea et al., 2004), where the mesh moves in the vertical direction. The ALE formulation guarantees strict local and global conservation of volume and tracers and allows for the use of generic vertical grids (Petersen et al., 2015).

All numerical ocean models include some form of friction, either in the form of a numerical closure or a physical parametrization (Griffies and Hallberg, 2000). Numerical closure involves adding a sufficient amount of dissipation to maintain numerical stability. There is a wealth of literature about stable finite volume (e.g., Danilov, 2012) and finite element discretizations (e.g., Hanert et al., 2003; Cotter et al., 2009a, b; Comblen et al., 2010b; McRae and Cotter, 2014) for rotational shallow water equations. Most of these schemes are stable for external gravity waves, and hence do not require any additional dissipation.

Solving the 3D hydrostatic equations under strong baroclinic forcing, however, generates noise at the grid-scale that does require dampening. A common approach is to add some form of viscosity proportional to the grid Reynolds number (Griffies and Hallberg, 2000; Ilıcak et al., 2012). Griffies and Hallberg (2000) argue that conventional Laplacian viscosity has too wide





a spectrum and tends to dissipate physically relevant (larger) scales too much. They show that biharmonic viscosity dissipates smaller scales more, and is thus more appropriate for removing noise at the grid-scale. In contrast to numerical closures, physical parametrizations aim to represent unresolved sub-grid-scale processes, such as strong lateral mixing near coasts or mixing at the bottom boundary layers. In this article, we focus on numerical closures; the presented viscosity schemes are mostly

motivated by numerical stability considerations.

In this article, we present an efficient DG implementation of the three-dimensional hydrostatic equations. The model is implemented in the *Thetis* project – an open source coastal ocean circulation model freely available online (see thetisproject.org). Thetis implements both a 2D depth-averaged circulation model and a full 3D hydrostatic model, the latter of which is discussed herein.

Thetis is implemented using the Firedrake finite element modeling platform (www.firedrakeproject.org; Rathgeber et al., 2016). We have chosen Firedrake because of its flexibility, and support for extruded meshes (McRae et al., 2016; Bercea et al., 2016). Firedrake uses high-level abstractions for describing the weak formulation of partial differential equations, specifically the Unified Form Language (Alnæs et al., 2014), and automated code generation to produce efficient C code (Homolya et al., 2017; Luporini et al., 2017) and just-in-time compilation. As such it is an extremely flexible modeling framework that does not

sacrifice computational efficiency; it is also an ideal platform for experimenting and benchmarking different discretizations. Automated code generation can also support different target hardware architectures, making it attractive for current and emerging high-performance computing platforms. In addition, Firedrake can automatically derive the adjoint of the forward model (Farrell et al., 2013), permitting inverse modeling applications such as parameter optimization and data assimilation.

The governing equations are presented in Section 2, followed by their DG finite element discretization in Section 3. The

20 second-order coupled time integration scheme is described in Section 4. Numerical tests are presented in Section 5.

## 2 Governing equations

Let $\Omega$ be the three-dimensional domain that spans from the sea floor $z = -h(x,y)$ to the free surface $z = \eta(x,y)$; the bottom and top surfaces are denoted by $\Gamma_b$ and $\Gamma_s$, respectively. Total water column depth is thus $H = \eta + h$. The two-dimensional horizontal domain is denoted by $\Gamma_0$.

The horizontal momentum equation reads

$$
\begin{aligned}
&\frac{\partial \boldsymbol{u}}{\partial t} + \boldsymbol{\nabla}_h \cdot (\boldsymbol{u}\boldsymbol{u}) + \frac{\partial (w\boldsymbol{u})}{\partial z} + f\boldsymbol{e}_z \wedge \boldsymbol{u} + \frac{1}{\rho_0}\boldsymbol{\nabla}_h p \\
&= \boldsymbol{\nabla}_h \cdot (\nu_h \boldsymbol{\nabla}_h \boldsymbol{u}) + \frac{\partial}{\partial z}\left(\nu \frac{\partial \boldsymbol{u}}{\partial z}\right),
\end{aligned} \tag{1}
$$

where $\boldsymbol{u} = (u, v)$ and $w$ denote the horizontal and vertical velocity, respectively; $\boldsymbol{\nabla}_h$ is the horizontal gradient operator; $\wedge$ denotes the cross product operator; $f$ is the Coriolis parameter; $\boldsymbol{e}_z$ is the vertical unit vector; $p$ is the pressure; and $\nu_h$ and $\nu$

are the horizontal and vertical diffusivity, respectively. Water density is defined as $\rho = \rho_0 + \rho'(T, S, p)$, where $T, S$ stand for temperature and salinity, respectively, and $\rho_0$ is a constant reference density.



Under the hydrostatic assumption the horizontal pressure gradient can be written as a combination of external, internal, and atmospheric pressure gradients:

$$\frac{1}{\rho_0}\boldsymbol{\nabla}_h p = g\boldsymbol{\nabla}_h\eta + g\boldsymbol{\nabla}_h r + \frac{1}{\rho_0}\boldsymbol{\nabla}_h p_{\mathrm{atm}}, \tag{2}$$

where $p_{\mathrm{atm}}$ is the atmospheric pressure acting on the sea surface, and

$$r = \frac{1}{\rho_0}\int\limits_z^\eta \rho' dz' \tag{3}$$

is the baroclinic head. For brevity the internal pressure gradient field is denoted as $\boldsymbol{F}_{\mathrm{pg}} = g\boldsymbol{\nabla}_h r$.

Neglecting atmospheric pressure, the full horizontal momentum equation reads

$$\frac{\partial \boldsymbol{u}}{\partial t} + \boldsymbol{\nabla}_h\cdot(\boldsymbol{u}\boldsymbol{u}) + \frac{\partial(w\boldsymbol{u})}{\partial z} + f\boldsymbol{e}_z\wedge\boldsymbol{u} + g\boldsymbol{\nabla}_h\eta + \boldsymbol{F}_{\mathrm{pg}} = \boldsymbol{\nabla}_h\cdot(\nu_h\boldsymbol{\nabla}_h\boldsymbol{u}) + \frac{\partial}{\partial z}\left(\nu\frac{\partial \boldsymbol{u}}{\partial z}\right). \tag{4}$$

Vertical velocity $w$ is diagnosed from the continuity equation:

$$\boldsymbol{\nabla}_h\cdot\boldsymbol{u} + \frac{\partial w}{\partial z} = 0. \tag{5}$$

Water temperature and salinity are modeled with an advection-diffusion equation of the form

$$\frac{\partial T}{\partial t} + \boldsymbol{\nabla}_h\cdot(\boldsymbol{u}T) + \frac{\partial(wT)}{\partial z} = \boldsymbol{\nabla}_h\cdot(\mu_h\boldsymbol{\nabla}_h T) + \frac{\partial}{\partial z}\left(\mu\frac{\partial T}{\partial z}\right), \tag{6}$$

where $\mu_h, \mu$ stand for the horizontal and vertical (eddy) diffusivity, respectively.

At the bottom boundary we impose quadratic bottom stress

$$\left(\nu_h\boldsymbol{n}_h\cdot\boldsymbol{\nabla}_h\boldsymbol{u} + \nu n_z\frac{\partial \boldsymbol{u}}{\partial z}\right)\bigg|_{\boldsymbol{x}\in\Gamma_b} = \frac{\boldsymbol{\tau}_b}{\rho_0}, \tag{7}$$

$$\frac{\boldsymbol{\tau}_b}{\rho_0} = C_d|\boldsymbol{u}_{bf}|\boldsymbol{u}_{bf}, \tag{8}$$

where $C_d$ is the drag coefficient, and $\boldsymbol{u}_{bf}$ is the velocity in the middle of the bottommost element. $\boldsymbol{n} = (n_x, n_y, n_z)$ is the outward normal vector, and $\boldsymbol{n}_h = (n_x, n_y, 0)$ its horizontal projection. The bottom boundary condition is treated implicitly; (8) is linearized by keeping the magnitude $|\boldsymbol{u}_{bf}|$ fixed at the "old" value while solving for $\boldsymbol{u}$ (and $\boldsymbol{u}_{bf}$). Typically $C_d$ is computed from the logarithmic law of the wall (e.g. Kärnä et al., 2013).

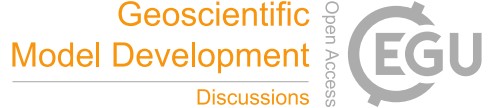



## 2.1 Mode splitting

Following Higdon and de Szoeke (1997) we split the horizontal velocity field into depth-averaged $\bar{\boldsymbol{u}}$ and deviation $\boldsymbol{u}' = \boldsymbol{u} - \bar{\boldsymbol{u}}$ components. The depth-averaged momentum equation is then defined as

$$\frac{\partial \bar{\boldsymbol{u}}}{\partial t} + f\boldsymbol{e}_z \wedge \bar{\boldsymbol{u}} + g\boldsymbol{\nabla}_h \eta = \boldsymbol{G}, \tag{9}$$

5  where $\boldsymbol{G}$ is a forcing term used to couple the 2D and 3D modes. This equation is complemented with the depth-averaged continuity (free surface) equation:

$$\frac{\partial \eta}{\partial t} + \boldsymbol{\nabla}_h \cdot (H\bar{\boldsymbol{u}}) = 0. \tag{10}$$

The 2D system (9)–(10) contains the fast-propagating, rotational surface gravity waves. The corresponding equation for $\boldsymbol{u}'$ is obtained by subtracting (9) from (4) (Higdon and de Szoeke, 1997):

$$\frac{\partial \boldsymbol{u}'}{\partial t} + \boldsymbol{\nabla}_h \cdot (\boldsymbol{u}\boldsymbol{u}) + \frac{\partial (w\boldsymbol{u})}{\partial z} + f\boldsymbol{e}_z \wedge \boldsymbol{u}' + \boldsymbol{F}_{\text{pg}}$$
$$= \boldsymbol{\nabla}_h \cdot (\nu_h \boldsymbol{\nabla}_h \boldsymbol{u}) + \frac{\partial}{\partial z}\left(\nu \frac{\partial \boldsymbol{u}}{\partial z}\right) - \boldsymbol{G}. \tag{11}$$

Note that the advection and viscosity terms are included in (11) without splitting, based on the assumption that these processes are slow enough to be captured with long time steps. The Coriolis term, on the other hand, only contains the slow modes. The vertical velocity $w$ only appears in the advection term, which is not split, and thus there is no need to split $w$.

## 15  2.2 Coupling 2D and 3D modes

The 2D and 3D modes are coupled using the additional term $\boldsymbol{G}$ (Higdon and de Szoeke, 1997; Ringler et al., 2013). First, the 3D momentum equation (11) is solved with $\boldsymbol{G} = 0$, resulting in a velocity field $\boldsymbol{u}'$ that has a non-zero depth-average, generated by the advection and viscosity terms (that depend on $\bar{\boldsymbol{u}}$). We then compute the depth-average $\overline{\boldsymbol{u}'}$ and apply a correction:

$$\boldsymbol{G} = \overline{\boldsymbol{u}'}/\Delta t, \tag{12}$$

20  $$\boldsymbol{u}' \leftarrow \boldsymbol{u}' - \boldsymbol{G}\Delta t \tag{13}$$

to enforce zero depth-average. By definition, the field $\boldsymbol{G}$ is a constant over the vertical, and it will be used as a forcing term in the 2D momentum equation (9) in the subsequent solve. This procedure ensures that equations (9) and (11) sum up to (4) and $\int \boldsymbol{u}' dz = 0$.





## 2.3 Equation of state

In this paper a linear equation of state is used:

$$\rho(T,S) = \rho_0 - \alpha_T(T - T_0) + \beta_S(S - S_0), \tag{14}$$

where $\alpha_T, \beta_S$ are the thermal expansion and saline contraction coefficients, respectively, and $T_0, S_0$ are reference temperature
and salinity. In all the test cases presented herein salinity does not contribute to water density ($\beta_S = 0$). Thetis also implements
a full non-linear equation of state (Jackett et al., 2006).

## 2.4 Viscosity and turbulence closure

Baroclinic flows require some form of viscosity to filter out grid-scale noise. In this paper we only consider Laplacian horizontal
viscosity, set to a constant $\nu_h = U\Delta x/\mathrm{Re}_h$ corresponding to the velocity scale $U$, horizontal mesh resolution $\Delta x$, and the
desired grid Reynolds number $\mathrm{Re}_h$. Here the velocity scale $U$ is taken as a global constant specific to each test case. Unless
otherwise specified, the horizontal diffusivity of tracers is zero.

In most test cases vertical viscosity is set to a constant. In certain cases we use the gradient Richardson number dependent
parametrization by Pacanowski and Philander (1981):

$$
\begin{aligned}
\nu &= \frac{\nu_0}{(1 + \alpha\mathrm{Ri})^n} + \nu_b, \\
\mu &= \frac{\nu}{1 + \alpha\mathrm{Ri}} + \mu_b,
\end{aligned}
\tag{15}
$$

where $\mathrm{Ri} = N^2/M^2$ is the gradient Richardson number, $N$ is the buoyancy frequency, and $M$ is the vertical shear frequency.
The background values are set to $\nu_b = \mu_b = 2 \times 10^{-5}$ m$^2$s$^{-1}$, while maximum viscosity is set to $\nu_0 = 2 \times 10^{-2}$ m$^2$s$^{-1}$; the
dimensionless parameters are $\alpha = 10$ and $n = 2$ (Wang et al., 2008b). More sophisticated turbulence closures will be addressed
in future work.

## 3 Finite element discretization

This section describes the spatial discretization of the governing equations. In Section 3.1 we define the finite element function
spaces, followed by the weak forms of the underlying equations.

### 3.1 Function spaces

The prognostic variables of the coupled 2D–3D system (9,10,11,6) are $\eta, \bar{\boldsymbol{u}}, \boldsymbol{u}', T$, and $S$. Diagnostic variables include the
vertical velocity $w$, water density $\rho'$, baroclinic head $r$, and internal pressure gradient $\boldsymbol{F}_{\mathrm{pg}}$. The choice of function spaces where
these variables reside is crucial for numerical stability and accuracy.

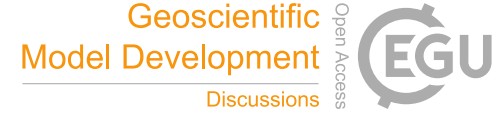



| Prognostic variables | | | |
|---|---|---|---|
| Field | Symbol | Equation | Function space |
| Water elevation | $\eta$ | (23) | $P_1^{DG}$ |
| Depth av. velocity | $\bar{\boldsymbol{u}}$ | (24) | $[P_1^{DG}]^2$ |
| Horizontal velocity | $\boldsymbol{u}'$ | (25) | $[P_1^{DG} \times P_1^{DG}]^2$ |
| Water temperature | $T$ | (26) | $P_1^{DG} \times P_1^{DG}$ |
| Water salinity | $S$ | (26) | $P_1^{DG} \times P_1^{DG}$ |
| Diagnostic variables | | | |
| Field | Symbol | Equation | Function space |
| Vertical velocity | $w$ | (31) | $P_1^{DG} \times P_1^{DG}$ |
| Water density | $\rho'$ | (14) | $P_1^{DG} \times P_1^{DG}$ |
| Baroclinic head | $r$ | (32) | $P_1^{DG} \times P_2$ |
| Int. pressure grad. | $\boldsymbol{F}_{pg}$ | (33) | $[P_1^{DG} \times P_1^{DG}]^2$ |

**Table 1.** Prognostic and diagnostic variables and their function spaces.

Our discretization is based on the linear discontinuous Galerkin function space, $P_1^{DG}$. The 2D system is discretized with a $P_1^{DG} - P_1^{DG}$ velocity–pressure finite element pair: Water elevation and both components of the depth-averaged velocity are approximated in $P_1^{DG}$ space, i.e. $\eta \in \mathcal{H}_{2D} = P_1^{DG}$, $\boldsymbol{u} \in \mathcal{U}_{2D} = [P_1^{DG}]^2$. When embedded with appropriate Riemann fluxes at element interfaces the $P_1^{DG} - P_1^{DG}$ element pair is well suited for rotational shallow water problems (Comblen et al., 2010b; Kärnä et al.,

2011).

Achieving an accurate and monotone 3D tracer advection scheme is one of our main design criteria. The tracers therefore are also considered within a discontinuous function space, $T, S \in \mathcal{H} = P_1^{DG} \times P_1^{DG}$ (here the $\times$ operator stands for the Cartesian product of function spaces in the extruded mesh: horizontal $\times$ vertical function space). Tracer consistency (sometimes called local tracer conservation) is a necessary condition for monotonicity; it ensures that a constant tracer field does not exhibit

spurious local extrema. In practice it implies that the discrete tracer equation must reduce to the discrete continuity equation for a constant tracer. In this work we satisfy this property by requiring that the vertical velocity belongs to the tracer space $\mathcal{H}$ (White et al., 2008). In addition, compatibility between the 2D and 3D momentum equations requires that the 3D horizontal velocity must be $P_1^{DG}$ in the horizontal direction. We therefore set $\boldsymbol{u}' \in \mathcal{U} = [P_1^{DG} \times P_1^{DG}]^2$ as well.

Note that this choice of function spaces is not *mimetic* (McRae and Cotter, 2014; Danilov, 2013): the discrete system does

not preserve all the properties of the continuous equations, for example enstrophy is not conserved exactly. As the coastal ocean is generally very dissipative, maintaining mimetic properties is however not crucial. It is possible to define a mimetic discretization as well, for example using Raviart-Thomas elements for the velocity, i.e. element pair $RT_1 - P_1^{DG}$ (McRae and Cotter, 2014). Our preliminary experiments however indicate that this choice significantly increases the computational cost of the system, without a corresponding improvement in accuracy. Formal assessment of the performance of mimetic discretiza-

tions in coastal ocean applications will be investigated in the future.





In the weak forms we use the following notation for volume and interface integrals

$$\left\langle \bullet \right\rangle_\Omega = \int_\Omega \bullet \, \mathrm{d}\boldsymbol{x}, \tag{16}$$

$$\left\langle\!\left\langle \bullet \right\rangle\!\right\rangle_{\partial\Omega} = \int_{\partial\Omega} \bullet \, \mathrm{d}s. \tag{17}$$

In interface terms we additionally use the average $\{\!\{\cdot\}\!\}$ and jump $[\![\cdot]\!]$ operators for scalar $a$ and vector $\boldsymbol{u}$ fields:

$$\{\!\{a\}\!\} = \frac{1}{2}(a^+ + a^-), \tag{18}$$

$$\{\!\{\boldsymbol{u}\}\!\} = \frac{1}{2}(\boldsymbol{u}^+ + \boldsymbol{u}^-), \tag{19}$$

$$[\![a\boldsymbol{n}]\!] = a^+\boldsymbol{n}^+ + a^-\boldsymbol{n}^-, \tag{20}$$

$$[\![\boldsymbol{u}\cdot\boldsymbol{n}]\!] = \boldsymbol{u}^+\cdot\boldsymbol{n}^+ + \boldsymbol{u}^-\cdot\boldsymbol{n}^-, \tag{21}$$

$$[\![\boldsymbol{u}\boldsymbol{n}]\!] = \boldsymbol{u}^+\boldsymbol{n}^+ + \boldsymbol{u}^-\boldsymbol{n}^-, \tag{22}$$

where the superscripts '+' and '−' arbitrarily label the values on either side of the interface and $\boldsymbol{n}$ is the outward unit normal vector of each element on the interface.

### 3.2 2D system

Let $\mathcal{T}$ stand for the triangulation of the 2D domain $\Gamma_0$. The set of element interfaces are denoted by $\mathcal{I} = \{k \cap k' | k, k' \in \mathcal{T}\}$, and $\boldsymbol{n} = (n_x, n_y)$ the outward unit normal vector of an interface $e \in \mathcal{I}$. For brevity boundary conditions are omitted from the weak forms.

Let $\phi_{2\mathrm{D}} \in \mathcal{H}_{2\mathrm{D}}$ and $\boldsymbol{\psi}_{2\mathrm{D}} \in \mathcal{U}_{2\mathrm{D}}$ be test functions in the 2D function spaces. The weak formulation of the 2D system then reads, find $\eta \in \mathcal{H}_{2\mathrm{D}}, \bar{\boldsymbol{u}} \in \mathcal{U}_{2\mathrm{D}}$ such that

$$\left\langle \frac{\partial\eta}{\partial t}\phi_{2\mathrm{D}} \right\rangle_{\Gamma_0} + \left\langle\!\left\langle (H^*\bar{\boldsymbol{u}}^*)\cdot[\![\phi_{2\mathrm{D}}\boldsymbol{n}]\!] \right\rangle\!\right\rangle_{\mathcal{I}} - \left\langle (H\bar{\boldsymbol{u}})\cdot\boldsymbol{\nabla}_h\phi_{2\mathrm{D}} \right\rangle_{\Gamma_0} = 0, \tag{23}$$

$$\left\langle \frac{\partial\bar{\boldsymbol{u}}}{\partial t}\cdot\boldsymbol{\psi}_{2\mathrm{D}} \right\rangle_{\Gamma_0} + \left\langle f\boldsymbol{e}_z \wedge \bar{\boldsymbol{u}}\cdot\boldsymbol{\psi}_{2\mathrm{D}} \right\rangle_{\Gamma_0} + \left\langle\!\left\langle g\eta^*[\![\boldsymbol{\psi}_{2\mathrm{D}}\cdot\boldsymbol{n}]\!] \right\rangle\!\right\rangle_{\mathcal{I}} - \left\langle g\eta\boldsymbol{\nabla}_h\cdot\boldsymbol{\psi}_{2\mathrm{D}} \right\rangle_{\Gamma_0} = \left\langle \boldsymbol{G}\cdot\boldsymbol{\psi}_{2\mathrm{D}} \right\rangle_{\Gamma_0}, \quad \forall\phi_{2\mathrm{D}} \in \mathcal{H}_{2\mathrm{D}}, \boldsymbol{\psi}_{2\mathrm{D}} \in \mathcal{U}_{2\mathrm{D}}. \tag{24}$$

Here the divergence $\boldsymbol{\nabla}_h\cdot(H\bar{\boldsymbol{u}})$ and external gradient $g\boldsymbol{\nabla}_h\eta$ terms have been integrated by parts. The resulting interface terms are defined on the element edges where the state variables $\eta, \bar{\boldsymbol{u}}$ are not uniquely defined. The values $\eta^*, \bar{\boldsymbol{u}}^*$ are obtained from an approximate Riemann solver; here we use the linear Roe solution $\eta^* = \{\!\{\eta\}\!\} + \sqrt{H/g}[\![\bar{\boldsymbol{u}}\cdot\boldsymbol{n}]\!]$ and $\bar{\boldsymbol{u}}^* = \{\!\{\bar{\boldsymbol{u}}\}\!\} + \sqrt{g/H}[\![\eta\boldsymbol{n}]\!]$ (Comblen et al., 2010b).



### 3.3 Momentum equation

Let $\mathcal{P}$ denote the set of prisms of the 3D domain $\Omega$, obtained from a vertical extrusion of $\Gamma_0$. The set of horizontal and vertical interfaces are denoted by $\mathcal{I}_h$ and $\mathcal{I}_v$, respectively. Let $\boldsymbol{\psi} \in \mathcal{U}$ be a test function. The weak formulation of the 3D momentum equation then reads: find $\boldsymbol{u} \in \mathcal{U}$ such that

$$
\begin{aligned}
\quad & \left\langle \frac{\partial \boldsymbol{u}'}{\partial t} \cdot \boldsymbol{\psi} \right\rangle_\Omega - \left\langle \boldsymbol{\nabla}_h \boldsymbol{\psi} : (\boldsymbol{u}\boldsymbol{u}) \right\rangle_\Omega + \left\langle\!\!\left\langle \boldsymbol{u}^{\mathrm{up}} \cdot [\![\boldsymbol{\psi}\boldsymbol{n}_h]\!] \cdot \{\!\!\{\boldsymbol{u}\}\!\!\} \right\rangle\!\!\right\rangle_{\mathcal{I}_h \cup \mathcal{I}_v} + \left\langle\!\!\left\langle \gamma_{\mathrm{lf}} [\![\boldsymbol{u}]\!] \cdot [\![\boldsymbol{\psi}]\!] \right\rangle\!\!\right\rangle_{\mathcal{I}_h \cup \mathcal{I}_v} \\
& - \left\langle (w\boldsymbol{u}) \cdot \frac{\partial \boldsymbol{\psi}}{\partial z} \right\rangle_\Omega + \left\langle\!\!\left\langle \boldsymbol{u}^{\mathrm{up}} \cdot [\![\boldsymbol{\psi} n_z]\!] \{\!\!\{w\}\!\!\} \right\rangle\!\!\right\rangle_{\mathcal{I}_h} \\
& + \left\langle f\boldsymbol{e}_z \wedge \boldsymbol{u}' \cdot \boldsymbol{\psi} \right\rangle_\Omega + \left\langle \boldsymbol{F}_{\mathrm{pg}} \cdot \boldsymbol{\psi} \right\rangle_\Omega = D_h(\boldsymbol{u}, \boldsymbol{\psi}) + D_v(\boldsymbol{u}, \boldsymbol{\psi}), \; \forall \boldsymbol{\psi} \in \mathcal{U}.
\end{aligned}
\tag{25}
$$

Here the advection and viscosity terms have been integrated by parts (see Kärnä et al., 2013); the colon operator is the Frobenius inner product, $\boldsymbol{A} : \boldsymbol{B} = \sum_{i,j} A_{i,j} B_{i,j}$, and $\boldsymbol{u}^{\mathrm{up}}$ stands for the upwind value at the interface. The horizontal advection operator has been augmented with the Lax-Friedrichs flux with parameter $\gamma_{\mathrm{lf}} = \{\!\!\{|\boldsymbol{u}|\}\!\!\}$. Adding such a flux in the momentum equation

reduces noise in the velocity field, and decreases spurious numerical mixing in baroclinic applications. The $D_h, D_v$ terms denote the diffusion operators introduced later.

### 3.4 Tracer equation

The weak formulation of the tracer equations is derived analogously: find $T \in \mathcal{H}$ such that

$$
\begin{aligned}
& \left\langle \frac{\partial T}{\partial t} \phi \right\rangle_\Omega - \left\langle T\boldsymbol{u} \cdot \boldsymbol{\nabla}_h \phi \right\rangle_\Omega + \left\langle\!\!\left\langle T^{\mathrm{up}} [\![\phi \boldsymbol{n}_h]\!] \cdot \{\!\!\{\boldsymbol{u}\}\!\!\} \right\rangle\!\!\right\rangle_{\mathcal{I}_h \cup \mathcal{I}_v} + \left\langle\!\!\left\langle \gamma_{\mathrm{lf}} [\![T]\!] [\![\phi]\!] \right\rangle\!\!\right\rangle_{\mathcal{I}_h \cup \mathcal{I}_v} \\
\quad & - \left\langle (Tw) \frac{\partial \phi}{\partial z} \right\rangle_\Omega + \left\langle\!\!\left\langle T^{\mathrm{up}} [\![\phi n_z]\!] \{\!\!\{w\}\!\!\} \right\rangle\!\!\right\rangle_{\mathcal{I}_v} = D_h(T, \phi) + D_v(T, \phi), \; \forall \phi \in \mathcal{H}.
\end{aligned}
\tag{26}
$$

Here the horizontal advection operator again uses Lax-Friedrichs flux.

### 3.5 Symmetric Interior Penalty stabilization

The presented discretization is unstable for elliptic operators, and the diffusion operators require additional stabilization. Here we use the Symmetric Interior Penalty Galerkin (SIPG) method (Epshteyn and Rivière, 2007). The SIPG formulation of the

20 tracer diffusion operators read



$$D_h(T,\phi) = -\left\langle \mu_h(\boldsymbol{\nabla}_h\phi)\cdot(\boldsymbol{\nabla}_h T)\right\rangle_\Omega + \left\langle\!\!\left\langle \{\!\!\{\mu_h\boldsymbol{\nabla}_h T\}\!\!\}\cdot[\![\phi\boldsymbol{n}_h]\!]\right\rangle\!\!\right\rangle_{\mathcal{I}_h\cup\mathcal{I}_v}$$
$$+ \left\langle\!\!\left\langle \{\!\!\{\mu_h\boldsymbol{\nabla}_h\phi\}\!\!\}\cdot[\![T\boldsymbol{n}_h]\!]\right\rangle\!\!\right\rangle_{\mathcal{I}_h\cup\mathcal{I}_v} - \left\langle\!\!\left\langle \{\!\!\{\sigma\}\!\!\}\{\!\!\{\mu_h\}\!\!\}[\![T\boldsymbol{n}_h]\!]\cdot[\![\phi\boldsymbol{n}_h]\!]\right\rangle\!\!\right\rangle_{\mathcal{I}_h\cup\mathcal{I}_v}, \tag{27}$$

$$D_v(T,\phi) = -\left\langle \mu\frac{\partial T}{\partial z}\frac{\partial\phi}{\partial z}\right\rangle_\Omega + \left\langle\!\!\left\langle \left\{\!\!\left\{\mu\frac{\partial T}{\partial z}\right\}\!\!\right\}[\![\phi n_z]\!]\right\rangle\!\!\right\rangle_{\mathcal{I}_h}$$
$$+ \left\langle\!\!\left\langle \left\{\!\!\left\{\mu\frac{\partial\phi}{\partial z}\right\}\!\!\right\}[\![T n_z]\!]\right\rangle\!\!\right\rangle_{\mathcal{I}_h} - \left\langle\!\!\left\langle \{\!\!\{\sigma\}\!\!\}\{\!\!\{\mu\}\!\!\}[\![T n_z]\!][\![\phi n_z]\!]\right\rangle\!\!\right\rangle_{\mathcal{I}_h}. \tag{28}$$

For the viscosity terms we get

$$D_h(\boldsymbol{u},\boldsymbol{\psi}) = -\left\langle \nu_h(\boldsymbol{\nabla}_h\boldsymbol{\psi}):(\boldsymbol{\nabla}_h\boldsymbol{u})^T\right\rangle_\Omega + \left\langle\!\!\left\langle [\![\boldsymbol{\psi}\boldsymbol{n}_h]\!]\cdot\{\!\!\{\nu_h\boldsymbol{\nabla}_h\boldsymbol{u}\}\!\!\}\right\rangle\!\!\right\rangle_{\mathcal{I}_h\cup\mathcal{I}_v}$$
$$+ \left\langle\!\!\left\langle [\![\boldsymbol{u}\boldsymbol{n}_h]\!]\cdot\{\!\!\{\nu_h\boldsymbol{\nabla}_h\boldsymbol{\psi}\}\!\!\}\right\rangle\!\!\right\rangle_{\mathcal{I}_h\cup\mathcal{I}_v} - \left\langle\!\!\left\langle \{\!\!\{\sigma\}\!\!\}\{\!\!\{\nu_h\}\!\!\}[\![\boldsymbol{u}\boldsymbol{n}_h]\!][\![\boldsymbol{\psi}\boldsymbol{n}_h]\!]\right\rangle\!\!\right\rangle_{\mathcal{I}_h\cup\mathcal{I}_v}, \tag{29}$$

$$D_v(\boldsymbol{u},\boldsymbol{\psi}) = -\left\langle \nu\frac{\partial\boldsymbol{\psi}}{\partial z}\cdot\frac{\partial\boldsymbol{u}}{\partial z}\right\rangle_\Omega + \left\langle\!\!\left\langle [\![\boldsymbol{\psi}n_z]\!]\cdot\left\{\!\!\left\{\nu\frac{\partial\boldsymbol{u}}{\partial z}\right\}\!\!\right\}\right\rangle\!\!\right\rangle_{\mathcal{I}_h}$$
$$+ \left\langle\!\!\left\langle [\![\boldsymbol{u}n_z]\!]\cdot\left\{\!\!\left\{\nu\frac{\partial\boldsymbol{\psi}}{\partial z}\right\}\!\!\right\}\right\rangle\!\!\right\rangle_{\mathcal{I}_h} - \left\langle\!\!\left\langle \{\!\!\{\sigma\}\!\!\}\{\!\!\{\nu\}\!\!\}[\![\boldsymbol{u}n_z]\!]\cdot[\![\boldsymbol{\psi}n_z]\!]\right\rangle\!\!\right\rangle_{\mathcal{I}_h}. \tag{30}$$

The penalty factor $\sigma$ is defined as $\sigma = \gamma\frac{p(p+1)}{L}$ (Epshteyn and Rivière, 2007), where $p$ is the degree of the basis functions, $\gamma$ is a factor depending on mesh quality, and $L$ is the local element length scale in the normal direction of the interface. Let $h_h$ and $h_v$ denote the horizontal and vertical element sizes, and $\boldsymbol{\Delta} = \text{diag}(h_h, h_h, h_v)$. We then define $L = \boldsymbol{n}\cdot\boldsymbol{\Delta}\cdot\boldsymbol{n} = h_h(n_x^2+n_y^2)+h_v n_z^2$ (Pestiaux et al., 2014). In this paper we use $\gamma = 5$.

### 3.6 Continuity equation

The vertical velocity $w$ is computed diagnostically from the continuity equation (5) by solving the weak form: find $w \in \mathcal{H}$ such that

$$\left\langle w n_z\varphi\right\rangle_{\Gamma_s} + \left\langle\!\!\left\langle \{\!\!\{w\}\!\!\}[\![\varphi n_z]\!]\right\rangle\!\!\right\rangle_{\mathcal{I}_h} - \left\langle w\frac{\partial\varphi}{\partial z}\right\rangle_\Omega = \left\langle \boldsymbol{u}\cdot\boldsymbol{\nabla}_h\varphi\right\rangle_\Omega - \left\langle\!\!\left\langle \{\!\!\{\boldsymbol{u}\}\!\!\}\cdot[\![\varphi\boldsymbol{n}_h]\!]\right\rangle\!\!\right\rangle_{\mathcal{I}_h\cup\mathcal{I}_v} - \left\langle\!\!\left\langle \boldsymbol{u}\cdot\varphi\boldsymbol{n}_h\right\rangle\!\!\right\rangle_{\Gamma_s}, \forall\varphi \in \mathcal{H}, \tag{31}$$

where both the left and right hand sides have been integrated by parts. Note that the terms on the bottom surface $\Gamma_b$ vanish due to the impermeability constraint $\boldsymbol{u}\cdot\boldsymbol{n}_h + w n_z = 0$.

### 3.7 Computing the internal pressure gradient

The water density is computed diagnostically using the equation of state. We use the same $P_1^{DG} \times P_1^{DG}$ function space for tracers and water density. In this work we use a linear equation of state (14), and consequently density can be computed locally (at each node of the tracer field). In general, however, the equation of state is non-linear, and the density is projected on the $\rho$ field.



The baroclinic head is computed from (3) by integrating $\rho'$ over the vertical. In practice we solve equation $\frac{\partial r}{\partial z} = \rho'/\rho_0$ weakly with the appropriate boundary conditions:

$$\left\langle r n_z \varphi \right\rangle_{\Gamma_b} + \left\langle\!\!\left\langle r_{\mathrm{up}} [\![\varphi n_z]\!] \right\rangle\!\!\right\rangle_{\mathcal{I}_h} - \left\langle r \frac{\partial \varphi}{\partial z} \right\rangle_\Omega = \left\langle \frac{1}{\rho_0} \rho' \varphi \right\rangle_\Omega. \tag{32}$$

Here the left hand side has been integrated by parts, and $r_{\mathrm{up}}$ denotes the value on the prism above the interface. Note that the free surface terms vanish because $r = 0$ on $\Gamma_s$ by definition. We use function space $\mathrm{P}_1^{\mathrm{DG}} \times \mathrm{P}_2$ for $r$ to alleviate internal pressure gradient errors (Piggott et al., 2008).

Finally, taking a test function $\boldsymbol{\psi} \in \mathcal{U}$, we compute the internal pressure gradient with the weak form

$$\left\langle \boldsymbol{F}_{\mathrm{pg}} \cdot \boldsymbol{\psi} \right\rangle_\Omega = -\left\langle g r \boldsymbol{\nabla}_h \cdot \boldsymbol{\psi} \right\rangle_\Omega + \left\langle\!\!\left\langle g \{\!\{r\}\!\} [\![\boldsymbol{\psi} \cdot \boldsymbol{n}_h]\!] \right\rangle\!\!\right\rangle_{\mathcal{I}_h \cup \mathcal{I}_v} + \left\langle\!\!\left\langle g r \boldsymbol{\psi} \cdot \boldsymbol{n}_h \right\rangle\!\!\right\rangle_{\Gamma_s \cup \Gamma_b}, \quad \forall \boldsymbol{\psi} \in \mathcal{U} \tag{33}$$

where the right hand side has been integrated by parts. Usually $\boldsymbol{F}_{\mathrm{pg}}$ belongs to the same space as the horizontal velocity, i.e. $[\mathrm{P}_1^{\mathrm{DG}} \times \mathrm{P}_1^{\mathrm{DG}}]^2$. However, to reduce bathymetry induced internal pressure gradient errors it is possible to use a quadratic horizontal space, i.e. $r \in \mathrm{P}_2^{\mathrm{DG}} \times \mathrm{P}_2$ and $\boldsymbol{F}_{\mathrm{pg}} \in [\mathrm{P}_2^{\mathrm{DG}} \times \mathrm{P}_1^{\mathrm{DG}}]^2$. In this paper we use a linear $\boldsymbol{F}_{\mathrm{pg}}$ field unless otherwise specified.

## 3.8 Slope limiters

We use vertex-based $\mathrm{P}_1^{\mathrm{DG}}$ slope limiters (Kuzmin, 2010) for three-dimensional variables to ensure positivity. The limiter is applied to both tracer and horizontal velocity fields after each update of the advection operator as discussed in the next Section.

## 4 Time integration

The coupled 2D–3D system is advanced in time with a two-stage arbitrary Lagrangian Eulerian (ALE) time integration scheme. In this section we present the ALE formulation and summarize the final time integration scheme.

### 4.1 ALE mesh formulation

To accurately account for the free surface movement one must move the mesh in the vertical direction. In this work we adopt the ALE method (Donea et al., 2004). Here we describe a mesh update procedure that stretches (or compresses) the mesh uniformly over the vertical direction. The ALE formulation, however, allows more complex mesh moving methods as well, such as the (approximate) tracking of isopycnals (Hofmeister et al., 2010).

In three dimensions an ALE update consists of solving an advection-diffusion equation between two domains, $\Omega^n$ and $\Omega^{n+1}$. Here the domain is uniquely defined by the surface elevation field, such that for any time level $n$ the surface $\Gamma_s^n$ matches $\eta^n$. Due to the chosen discretization the elevation field $\eta$ is discontinuous, yet we wish to maintain a conforming mesh, i.e. a continuous coordinate field $z$. This is achieved by projecting the elevation field $\eta^n$ to a continuous space and updating the geometry with the continuous field $\eta_{\mathrm{cg}}^n$. The projection induces a small discrepancy between the elevation field and the 3D domain, but its effect remains negligible in practical applications because jumps in the elevation field are typically small.





Let $\Omega^{\mathrm{ref}}$ be the reference domain corresponding to unperturbed elevation field $\eta_{\mathrm{cg}} = 0$, and $z_{\mathrm{ref}} \in [-h, 0]$ its vertical coordinate. Applying a uniform mesh stretching, the time dependent mesh coordinates can then be written as

$$z^n = z_{\mathrm{ref}} + \eta_{\mathrm{cg}}^n \frac{z_{\mathrm{ref}} + h}{h} \in [-h, \eta_{\mathrm{cg}}^n]. \tag{34}$$

The mesh velocity is obtained as $w_m = \frac{\partial z}{\partial t}$. In practice the consecutive fields $\eta_{\mathrm{cg}}^{n+1}$ and $\eta_{\mathrm{cg}}^n$ are known so we can evaluate

$$w_m^{n+1} = \frac{\eta_{\mathrm{cg}}^{n+1} - \eta_{\mathrm{cg}}^n}{\Delta t} \frac{z_{\mathrm{ref}} + h}{h}. \tag{35}$$

Given the mesh velocity a conservative ALE update can be written as

$$\frac{d}{dt} \left( \left\langle T\phi \right\rangle_{\Omega} \right) = \left\langle F_T(T, \boldsymbol{u}, w - w_m)\phi \right\rangle_{\Omega}, \tag{36}$$

for a generic tracer equation $\frac{\partial T}{\partial t} = F_T(T, \boldsymbol{u}, w)$.

## 4.2 Coupled time integration scheme

The coupled 2D–3D system is advanced in time with a two-stage ALE time integration scheme. For convenience we re-write the 3D momentum and tracer equations as

$$\frac{\partial T}{\partial t} = F_T(T, \boldsymbol{u}, w) + G_T(T), \tag{37}$$

$$\frac{\partial \boldsymbol{u}}{\partial t} = F_{\boldsymbol{u}}(\boldsymbol{F}_{\mathrm{pg}}, \boldsymbol{u}, w) + G_{\boldsymbol{u}}(\boldsymbol{u}), \tag{38}$$

where $F_T$ and $F_{\boldsymbol{u}}$ denote all the terms that are treated explicitly while $G_T$ and $G_{\boldsymbol{u}}$ contain all the implicit terms. In this work only vertical diffusion (28), vertical viscosity (30), and bottom friction terms are treated implicitly.

The explicit 3D equations are advanced in time with a second-order strong stability preserving (SSP) Runge-Kutta scheme, SSPRK(2,2) (Shu and Osher, 1988; Gottlieb and Shu, 1998). For a generic problem $\frac{\partial c}{\partial t} = F(c)$ the scheme reads:

$$c^{(1)} = c^n + \Delta t F(c^n), \tag{39}$$

$$c^{n+1} = c^n + \frac{1}{2}\Delta t F(c^n) + \frac{1}{2}\Delta t F(c^{(1)}). \tag{40}$$

When applied to the explicit 3D momentum and tracer equations, (25) and (26), both of these stages are ALE updates where the mesh is updated from geometry $\Omega^n$ to $\Omega^{(1)}$ and then $\Omega^{n+1}$. The ALE formulation of the explicit 3D tracer equation can



then be written as

$$\left\langle T^{(1)}\phi \right\rangle_{\Omega^{(1)}} = \left\langle T^n \phi \right\rangle_{\Omega^n} + \Delta t \left\langle F_T(T^n, \boldsymbol{u}^n, w^n - w_m^{(1)})\phi \right\rangle_{\Omega^n}, \tag{41}$$

$$\left\langle \widetilde{T}^{n+1}\phi \right\rangle_{\Omega^{n+1}} = \left\langle T^n \phi \right\rangle_{\Omega^n} + \frac{1}{2}\Delta t \left\langle F_T(T^n, \boldsymbol{u}^n, w^n - w_m^{(1)})\phi \right\rangle_{\Omega^n} +$$
$$\frac{1}{2}\Delta t \left\langle F_T(T^{(1)}, \boldsymbol{u}^{(1)}, w^{(1)} - w_m^{n+1})\phi \right\rangle_{\Omega^{(1)}}, \tag{42}$$

5   where the vertical velocity is adjusted by the mesh velocity $w_m$.

After the SSPRK update, the implicit terms are advanced with the backward Euler method. This step is computed in domain $\Omega^{n+1}$:

$$\left\langle T^{n+1}\phi \right\rangle_{\Omega^{n+1}} = \left\langle \widetilde{T}^{n+1}\phi \right\rangle_{\Omega^{n+1}} + \Delta t \left\langle G_T(T^{n+1})\phi \right\rangle_{\Omega^{n+1}}. \tag{43}$$

The 3D momentum equation is treated analogously.

---

**Algorithm 1** Summary of the coupled time integration algorithm.

---

**Require:** Model state variables at time $t_n$: $\eta^n, \bar{u}^n, T^n, S^n, \boldsymbol{u}'^n$

**First stage:**

1: Solve 2D system for $(\eta^{(1)}, \bar{u}^{(1)})$ (46)–(47)

2: Update mesh geometry to $\Omega^{(1)}$ and compute mesh velocity $w_m^{(1)}$ (35)

3: Update 3D equations with ALE step for $T^{(1)}, S^{(1)}, \boldsymbol{u}'^{(1)}$ (41)

4: Apply slope limiter to $T^{(1)}, S^{(1)}, \boldsymbol{u}'^{(1)}$

5: Update the 2D coupling term $\boldsymbol{G}$ (12) and correct $\boldsymbol{u}'$ (13)

6: Update $w$ (31), water density (14), and pressure gradient (33)

**Second stage:**

7: Solve 2D system for $(\eta^{n+1}, \bar{u}^{n+1})$ (48)–(49)

8: Update mesh geometry to $\Omega^{n+1}$ and compute mesh velocity $w_m^{n+1}$ (35)

9: Update 3D equations with ALE step for $\widetilde{T}^{n+1}, \widetilde{S}^{n+1}, \widetilde{\boldsymbol{u}}'^{n+1}$ (42)

10: Apply slope limiter to $\widetilde{T}^{n+1}, \widetilde{S}^{n+1}, \widetilde{\boldsymbol{u}}'^{n+1}$

**Final stage:**

11: Update the 2D coupling term $\boldsymbol{G}$ (12) and correct $\boldsymbol{u}'$ (13)

12: Solve vertical viscosity and diffusion implicitly (43)

13: Update $w$ (31), water density (14), and pressure gradient (33)

14: Update parametrizations (e.g. bottom friction and viscosity)

---

10   The 2D equations are advanced in time with an implicit scheme to circumvent the strict time step constraint imposed by surface gravity waves. To ensure consistency between the movement of the 3D mesh and the 2D mode, the 2D time integration scheme must be compatible with the aforementioned SSPRK(2,2) method. Here we use a combination of a forward Euler and trapezoidal steps:





$$c^{(1)} = c^n + \Delta t F(c^n), \tag{44}$$

$$c^{n+1} = c^n + \frac{1}{2}\Delta t \left( F(c^n) + F(c^{n+1}) \right). \tag{45}$$

Denoting the tendencies of the 2D system (23)-(24) by $F_\eta$ and $F_{\bar{u}}$, respectively, we can write the 2D solver as

$$\left\langle \eta^{(1)}\phi_{2D} \right\rangle_{\Gamma_0} = \left\langle \eta^n \phi_{2D} \right\rangle_{\Gamma_0} + \Delta t \left\langle F_\eta(H^n, \bar{u}^n)\phi_{2D} \right\rangle_{\Gamma_0}, \tag{46}$$

$$\left\langle \bar{\boldsymbol{u}}^{(1)} \cdot \boldsymbol{\psi}_{2D} \right\rangle_{\Gamma_0} = \left\langle \bar{\boldsymbol{u}}^n \cdot \boldsymbol{\psi}_{2D} \right\rangle_{\Gamma_0} + \Delta t \left\langle F_{\bar{u}}(\eta^n, \bar{u}^n) \cdot \boldsymbol{\psi}_{2D} \right\rangle_{\Gamma_0}, \tag{47}$$

$$\left\langle \eta^{n+1}\phi_{2D} \right\rangle_{\Gamma_0} = \left\langle \eta^n \phi_{2D} \right\rangle_{\Gamma_0} + \frac{\Delta t}{2}\left\langle \left( F_\eta(H^n, \bar{u}^n) + F_\eta(H^n, \bar{u}^{n+1}) \right)\phi_{2D} \right\rangle_{\Gamma_0}, \tag{48}$$

$$\left\langle \bar{\boldsymbol{u}}^{n+1} \cdot \boldsymbol{\psi}_{2D} \right\rangle_{\Gamma_0} = \left\langle \bar{\boldsymbol{u}}^n \cdot \boldsymbol{\psi}_{2D} \right\rangle_{\Gamma_0} + \frac{\Delta t}{2}\left\langle \left( F_{\bar{u}}(\eta^n, \bar{u}^n) + F_{\bar{u}}(\eta^{n+1}, \bar{u}^{n+1}) \right) \cdot \boldsymbol{\psi}_{2D} \right\rangle_{\Gamma_0}. \tag{49}$$

The second implicit stage is linearized by treating the total depth $H$ explicitly in (48).

The 2D system is solved first, resulting in an updated elevation field ($\eta^{(1)}$ and $\eta^{n+1}$ for the two stages, respectively) and consequently mesh geometry ($\Omega^{(1)}$ and $\Omega^{n+1}$). Once the mesh geometry is known it is straightforward to compute the corresponding mesh velocity $w_m$ and perform a 3D ALE update.

The time integration method is second-order for all the terms. The whole algorithm is summarized in Algorithm 1.

### 4.3 Choosing the time step

The maximal admissible time step is constrained by stability of the explicit solvers. The presented SSPRK(2,2) has a CFL (Courant–Friedrichs–Lewy) factor 1. Therefore the horizontal advection term imposes a constraint

$$\Delta t_{\text{adv}} = \frac{\sigma_h L_h}{U}, \tag{50}$$

where $L_h$ is the horizontal element size, $U$ is the maximal horizontal velocity scale, and $\sigma_h$ is a length scaling factor. For the presented $P_1^{\text{DG}}$ discretization we take $L_h$ as the square root of the triangle area and use a scaling factor $\sigma_h = 0.3125$, typical for DG methods (Rivière, 2008). In strongly stratified flows internal waves may impose a stricter constraint

$$\Delta t_{\text{iw}} = \frac{\sigma_h L_h}{C_{\text{iw}} + U}, \tag{51}$$

where $C_{\text{iw}}$ is the speed of the internal waves.

Analogously, the time step constraint for vertical advection is

$$\Delta t_{\text{w}} = \frac{\sigma_v L_z}{W}, \tag{52}$$

where $L_z$ is the element height, $W$ is the vertical velocity scale, and $\sigma_v = 0.125$ is the scaling factor.





Given a horizontal viscosity scale $N_h$, the explicit viscosity operator imposes a constraint

$$\Delta t_{\text{visc}} = \sigma_{\text{visc}} \frac{(\sigma_h L_h)^2}{N_h}. \tag{53}$$

which may become stringent for small elements and large viscosity values. The scaling factor $\sigma_{\text{visc}}$ depends on the used stabilization scheme; here a value $\sigma_{\text{visc}} = 2$ is used. The constraint for horizontal diffusivity is analogous.

In the simulations presented herein, the minimal admissible time step is evaluated on the mesh based on constant *a-priori* velocity and viscosity scales. The time step is kept constant throughout the simulation.

## 5   Test cases

We demonstrate the performance of the proposed discretization with a suite of test cases of increasing complexity. We first evaluate the conservation and convergence of the solver in a barotropic standing wave test case. The convergence of baroclinic

terms is then examined in a specific steady-state test case. The baroclinic solver and its numerical mixing is then evaluated with a non-rotating lock exchange test case and a rotating baroclinic eddy test, followed by the DOME overflow test.

### 5.1   Standing wave

We first evaluate the performance of the solver in a barotropic standing wave test case. The domain is a $L_x = 60$ km long rectangular channel, 625 m wide, and 100 m deep. All lateral boundaries are closed. Initially the water is at rest. A 10 m tall

sinusoidal elevation perturbation is prescribed along the channel ($\eta_a = -\eta_0 \cos(2\pi x/L_x)$, $\eta_0 = 10$ m), leading to a nonlinear wave as the simulation progresses.

    The simulation is run for two full wave periods, approximately 3831.31 s. To investigate tracer conservation and consistency properties two passive tracers are included: salinity is set to a constant 4.5 psu, while temperature varies between $5.0$ and $15.0\,^\circ$C along the channel ($T = 5\sin(2\pi x/L_x) + 10\,^\circ$C).

The domain was discretized with a split-quad mesh using 40 elements along the channel (1500 m edge length) and 4 vertical layers. The time step is $\Delta t = 95.78$ s, chosen to meet the horizontal advection condition.

    During the simulation the volume of the 3D domain was conserved to accuracy $\mathcal{O}(10^{-15})$. The "2D volume", i.e. the integral of the elevation field, was conserved to accuracy $\mathcal{O}(10^{-16})$. Salinity remained at constant 4.5 psu with a small $\mathcal{O}(10^{-9})$ deviation. The total mass of salinity and temperature were both conserved to accuracy $\mathcal{O}(10^{-12})$. Over- and undershoots in the

temperature field were negligible due to the slope limiters. Without the limiter temperature overshoots were $\mathcal{O}(10^{-2})$. These results show that the model indeed fully conserves volume and tracers and does not exhibit overshoots. Moreover, the tracer consistency property is satisfied, verifying the integrity of the ALE formulation.

    To investigate the order of convergence of the solver, we used a smaller initial elevation perturbation $\eta_0 = 1$ cm. In this case the resulting standing wave is close to linear. At the end of the simulation the solution was compared to the analytical solution

of the linear wave equation (which coincides with the initial condition) by computing the $L_2$ error, $\mathcal{E}(\eta) = (\int_\Omega (\eta - \eta_a)^2 \mathrm{d}\boldsymbol{x})^{1/2}$.





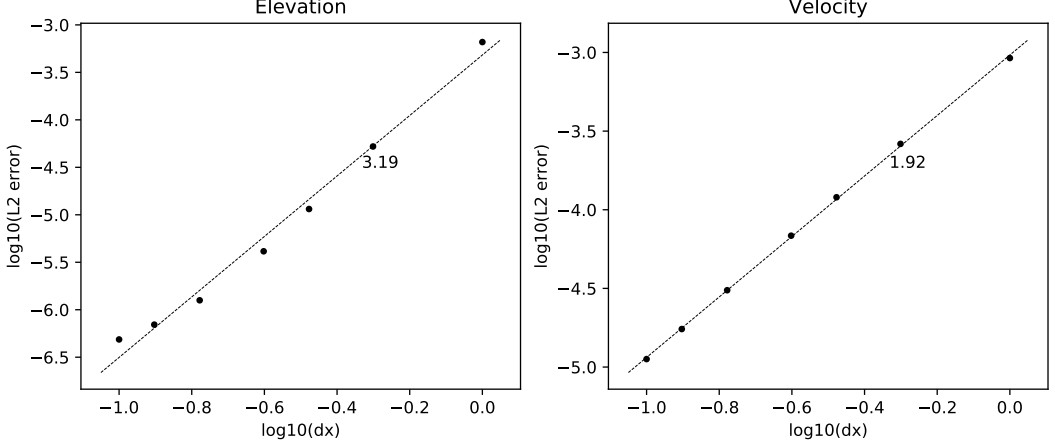

**Figure 1.** Convergence of the $L_2$ error in the standing wave test case. Tested element sizes were 3000, 1500, 1000, 750, 500, 375, and 300 m. The number indicates the slope of the least-squares best fit line (dashed line).

We ran the simulation varying the horizontal mesh resolution between 3 km and 300 m; the number of vertical levels varied between 2 and 20. In each case the channel was made one element wide, and the time step was chosen to meet the CFL criterion for horizontal advection. At the end of the simulation the $L_2$ error was computed for water elevation and velocity (see Figure 1). The velocity field shows the expected second-order convergence, whereas elevation converges at a rate of 3.2. It is known that

$P_1^{DG}$ shallow water equations models may exhibit superconvergence properties, especially for the elevation field (Bernard et al., 2008; Comblen et al., 2010b). Here our results verify that the solver behaves as expected and yields second-order accuracy under barotropic forcing.

## 5.2 Baroclinic MMS test

Verifying model accuracy under baroclinic forcing is more challenging as no analytical solutions are available. Here we use

the method of manufactured solutions (MMS; Salari and Knupp, 2000) to construct a steady state test case that allows us to verify the correctness of the discrete baroclinic operators. The domain is a $L_x = 15$ by $L_y = 10$ km large and $h = 40$ m deep rectangular box. All lateral boundaries are closed. We prescribe initial velocity and temperature fields

$$u_a = \frac{1}{2} \sin\left(\frac{2\pi}{L_x}x\right) \cos\left(\frac{3z}{h}\right), \tag{54}$$

$$v_a = \frac{1}{3} \cos\left(\frac{\pi y}{L_y}\right) \sin\left(\frac{z}{2h}\right), \tag{55}$$

$$T_a = 15 \sin\left(\frac{\pi x}{L_x}\right) \sin\left(\frac{\pi y}{L_y}\right) \cos\left(\frac{z}{h}\right) + 15. \tag{56}$$

These functions were chosen to be analytic (infinitely differentiable) and fully three-dimensional to better quantify the spatial discretization error.





Salinity is set to a constant 35 psu. We use the linear equation of state (14) with $\rho_0 = 1000$ kg m$^{-3}$, $\alpha_T = 0.2$ kg m$^{-3}$ °C$^{-1}$ and $T_0 = 5$ °C. For the sake of simplicity, bathymetry is constant and elevation is set to zero initially. Coriolis frequency was set to a constant $f = 10^{-4}$ s$^{-1}$. Bottom friction, viscosity, and diffusivity are omitted.

Without any additional forcing the initial conditions lead to a time-dependent solution. Following the MMS strategy, we add analytical source terms in the dynamic equations that cancel all the active terms in the equations, leading to a steady state solution. The remaining error is purely the discretization error of the advection, pressure gradient, and Coriolis operators. The source terms are derived analytically and projected to the corresponding function space. The analytical formulae are given in Appendix A.

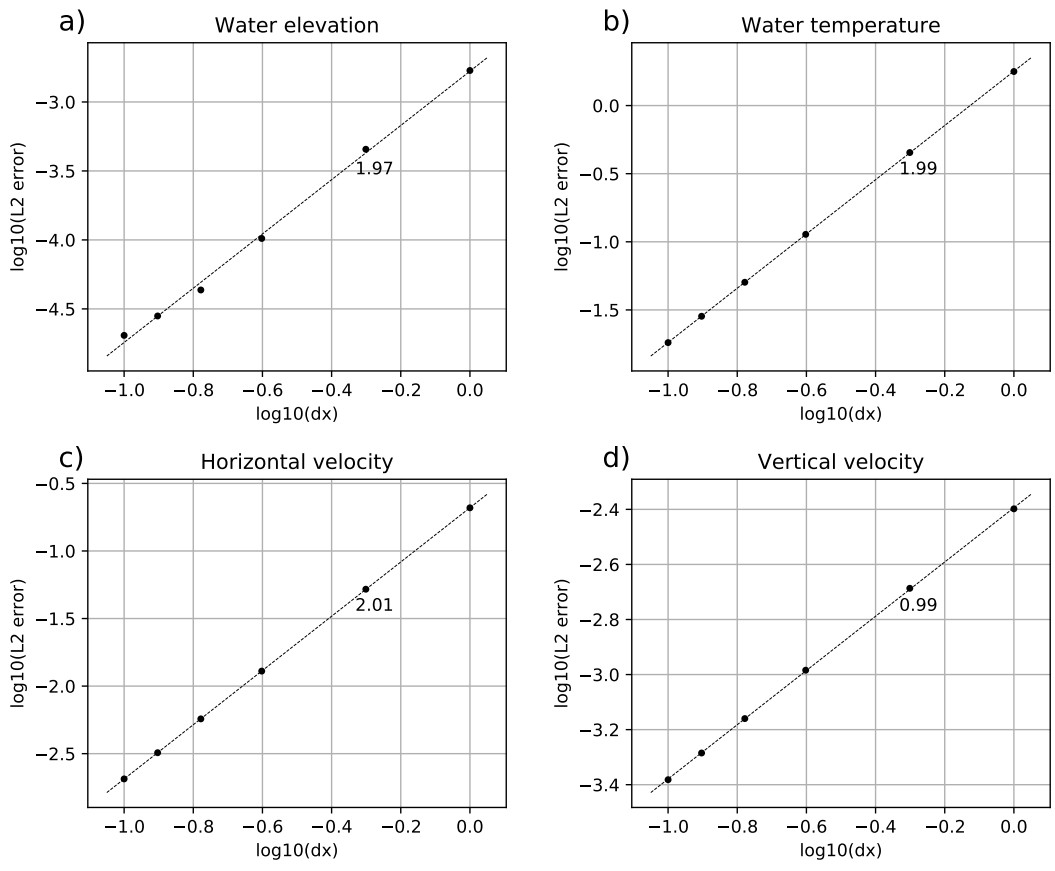

**Figure 2.** Convergence of the $L_2$ error in the baroclinic MMS test case. The mesh was refined 1, 2, 4, 6, 8, and 10 times, resulting in resolution 2500, 1250, 625, 416.67, 312.5, and 250 m (shortest edge of the triangle). The time step was 25.0, 12.5, 6.25, 4.167, 3.125, and 2.5 s, respectively. The number indicates the slope of the least-squares best fit line (dashed line).





The coarsest mesh contains 4 elements both in $x$ and $y$ directions and 2 vertical levels. We refine the mesh up to 10 times (40 elements and 20 vertical levels) and compute the $L_2$ error of the prognostic fields against the exact solutions. In each case, the model is run for 50 iterations with a time step chosen to meet the CFL condition.

The variation of the $L_2$ errors with resolution is shown in Figure 2. All the prognostic variables exhibit the correct second-order convergence rate. The diagnostic vertical velocity field (which depends on the divergence of $\boldsymbol{u}$) converges linearly as expected. Therefore we conclude that advection, pressure gradient, and Coriolis terms are discretized correctly. We have also developed similar MMS tests for the diffusivity/viscosity operators and the bottom friction term, all of which show second-order convergence as well (not shown).

### 5.3   Lock exchange

The validity of the baroclinic solver and its level of spurious mixing is investigated with the standard lock exchange test case (Wang, 1984; Haidvogel and Beckmann, 1999; Jankowski, 1999; Ilıcak et al., 2012; Kärnä et al., 2013; Petersen et al., 2015). Here we follow the setup of Ilıcak et al. (2012) and Petersen et al. (2015): The domain is a 64 km long and 1 km wide rectangular channel. Water depth is 20 m. Initially, the left-hand side of the domain is filled with dense water mass ($T = 5.0\,^{\circ}\mathrm{C}$) compared to the water on the right ($T = 30.0\,^{\circ}\mathrm{C}$). Salinity is kept at constant 35 psu. We use the same linear equation of state as in Section 5.2, resulting in a density difference of $\Delta\rho = 5.0$ kg m$^{-3}$. The domain is discretized with a regular split-quad mesh. The triangle edge length is 500 m and 20 equidistant $\sigma$ levels are used in the vertical direction.

Stabilizing the internal pressure gradient requires some form of friction. To this end, we apply a constant Laplacian horizontal viscosity, using values $\nu = 1.0,\ 10.0,\ 100.0$, and $200.0$ m$^2$s$^{-1}$. These values correspond to grid Reynolds numbers $\mathrm{Re}_h = U\Delta x/\nu = 250.0,\ 25.0,\ 2.5$, and $1.25$, respectively, where the characteristic velocity scale of the exchange flow is $U = 0.5$ m s$^{-1}$. Vertical viscosity is set to a constant $10^{-4}$ m$^2$s$^{-1}$. Bottom friction is omitted.

Figure 3 shows the initial density field and solution after 17 h of simulation for the three cases. Higher background viscosity leads to a less noisy velocity field and therefore sharper density front. The sharpness and shape of the fronts are similar to results presented in the literature (e.g. Fig. 5 in Ilıcak et al., 2012). The low viscosity cases ($\mathrm{Re}_h = 25, 250$) exhibit an internal wave at the front which significantly increases the overall mixing.

Assuming that, in the absence of bottom friction, all available potential energy is transformed into kinetic energy, the maximum front propagation speed can be estimated as $c = 1/2\sqrt{gH\Delta\rho/\rho_0}$ (Benjamin, 1968; Jankowski, 1999). Figure 4 (a) shows the propagation of the front location at the bottom of the domain (the front at the surface behaves comparably). All three simulations are in good agreement with the theoretical propagation speed. The simulated front propagation speed is underestimated by roughly 5% indicating loss of energy due to mixing. These results are similar to results reported in the literature; e.g. Ilıcak et al. (2012) show similar performance for ROMS, MITgcm, and MOM.

Figure 4 (b) shows the maximum over- and undershoots in the temperature field during the simulation. Even in the low viscosity case ($\mathrm{Re}_h = 250$), the overshoots are of order $10^{-5}\,^{\circ}\mathrm{C}$ indicating that the tracer advection scheme is indeed close to monotone, due to the SSP time integration method and slope limiters. Note that if the slope limiter is omitted, the overshoots can reach $30\,^{\circ}\mathrm{C}$.





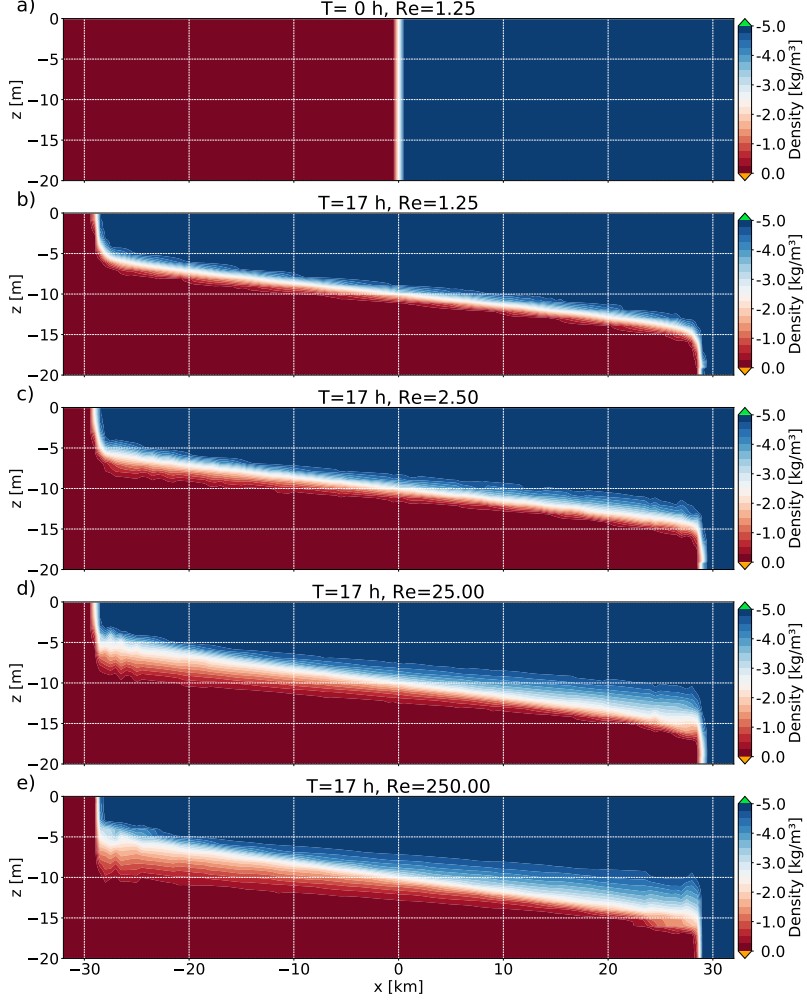

**Figure 3.** Water density in the lock exchange test case in the center of the domain ($y = 0$ km). (a) Initial condition. Solution after 17 h of simulation with Re$_h$ (b) 1.25, (c) 2.5, (d) 25.0, and (e) 250.0.

To diagnose the role of spurious mixing we use the reference potential energy (RPE, Ilıcak et al. 2012; Petersen et al. 2015). RPE is computed as the vertical center of mass of a sorted density field $\rho^*$: $RPE = g \int \rho^* (z + h) d\boldsymbol{x}$. The $\rho^*$ field is defined as the unique, stratified density field where the densest water parcels are distributed over the bottom, and density increases monotonically over the water column. As such, $\rho^*$ is the steady-state density distribution, and RPE represents the portion of potential energy that cannot be transformed into kinetic energy. Mixing the two water masses increases RPE (the center of mass) and thus the amount of unavailable potential energy increases. Figure 4 (c) shows the evolution of normalized RPE, $\overline{RPE}(t) = (RPE(t) - RPE(0))/RPE(0)$ during the simulation. At $t = 17$ h the values are $0.587, 1.14, 2.68, 3.42 \times 10^{-5}$ for the four simulations. These results are in good agreement with those reported with MPAS-Ocean model (Petersen et al.,





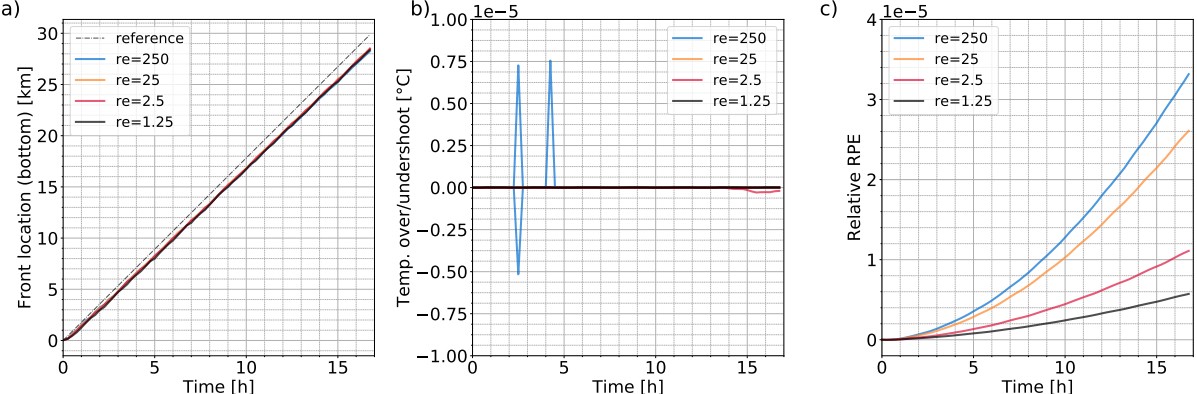

**Figure 4.** Diagnostics of the lock exchange test. (a) Location of the density front at the bottom of the domain, (b) Over- and undershoots in the temperature field (wrt. to 30.0 and 5.0 °C, respectively), (c) Normalized reference potential energy (RPE) versus simulation time.

2015): With the same mesh resolution, MPAS-Ocean shows slightly larger normalized RPE, for example, at $t = 17$ h $\overline{RPE} \approx 3.5 \times 10^{-5}$ in the case of $\mathrm{Re}_h = 25$. The difference is likely due to the different spatial discretization ($\mathrm{P}_1^{\mathrm{DG}}$ instead of finite volumes), or differences in the numerical viscosity operator. Applying slope limiters to the velocity field is not necessary for numerical stability, but it reduces high-frequency noise in the velocity field and hence results in lower RPE values.

## 5.4 Baroclinic eddies

We investigate the model's ability to generate baroclinic eddies with the eddying channel test case of Ilıcak et al. (2012). This test case is an idealization of the Antarctic Circumpolar Current, the domain spanning 500 km and 160 km in the meridional and zonal directions, respectively. The domain is 1000 m deep. At the zonal boundaries, periodic boundary conditions are applied; northern and southern boundaries are closed. The Coriolis parameter is taken as a constant $1.2 \times 10^{-4}$ s$^{-1}$.

Initially, the domain is linearly stratified with warmer water at the surface. In addition, the northern half of the domain is warmer, with a narrow sinusoidal transition band separating the warm (northern) and cold (southern) water masses (Figure 5; see Petersen et al. 2015 for the definition of the initial temperature field). Water temperature ranges between 10 and 20 °C. A linear equation of state is used with $\rho_0 = 1000$ kg m$^{-3}$, $\alpha_T = 0.2$ kg m$^{-3}$ °C$^{-1}$ and $T_0 = 5$ °C. Salinity is kept at constant 35 psu and it does not affect density ($\beta_S = 0$). Bottom friction is parametrized by a constant drag coefficient $C_D = 0.01$.

The baroclinic Rossby radius of deformation is 20 km (Ilıcak et al., 2012). Horizontal mesh resolution is constant in space. We use a regular split-quad mesh with two different mesh resolutions: eddy-permitting 10 km and a finer 4 km resolution. In the vertical direction, 26 and 40 equidistant sigma levels are used in the two cases, respectively. Simulations are carried out with different values of horizontal viscosity, the grid Reynolds number ranging from 2 to 100. The different setups are summarized in Table 2. Vertical viscosity is set to a constant $10^{-4}$ m$^2$s$^{-1}$.

As the simulation progresses, baroclinic eddies develop at the center of the domain, quickly propagating elsewhere. This is a spin-down experiment, i.e. the domain is a closed system with no forcing at the boundaries. Therefore all the energy in the





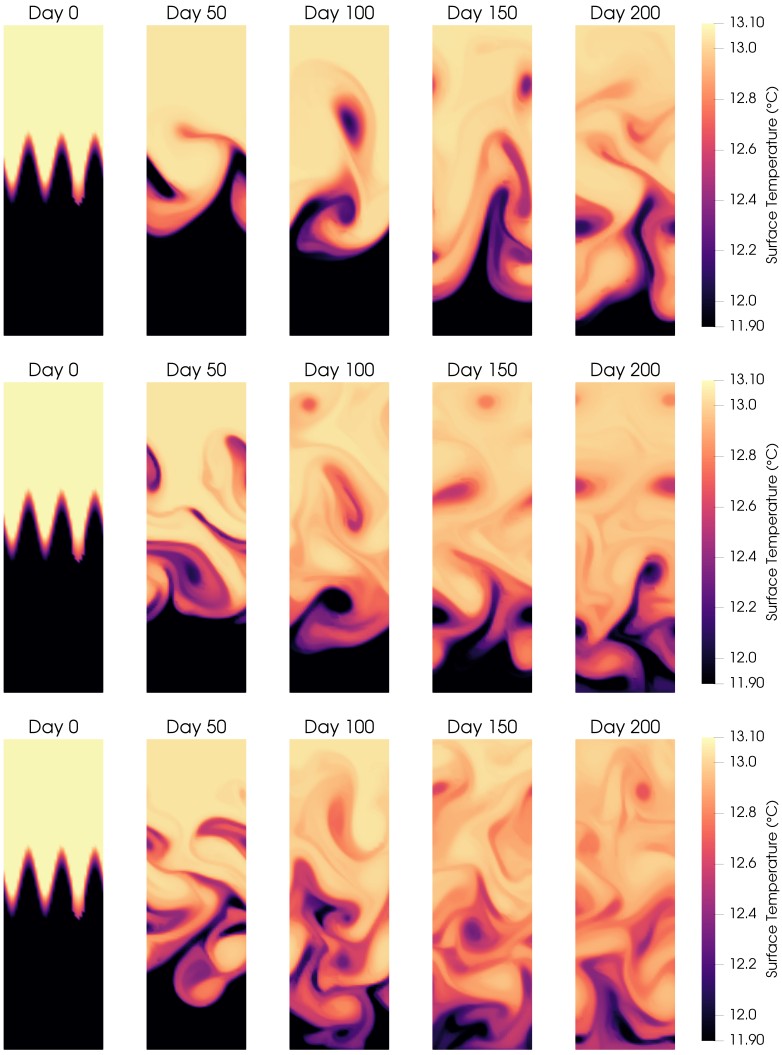

**Figure 5.** Sea surface temperature fields for the eddying channel test case at 4 km horizontal mesh resolution. Horizontal viscosity is 200 (top row), 50 (middle), and 20 $m^2 s^{-1}$ (bottom). These values correspond to mesh Reynolds numbers 2, 8, and 20, respectively.

system originates from the initial potential energy, which is being dissipated during the simulation; again the RPE is used as a metric for the energy transfer, or, the loss of energy due to mixing.

Figure 5 shows the surface temperature fields at various time intervals up to 200 days after the initialization for different values of horizontal viscosity. As expected, the model captures more mesoscale features as viscosity is decreased. Qualitatively, the results are in agreement with ROMS and MITgcm results (Ilıcak et al., 2012), as well as MPAS-Ocean (Petersen et al., 2015), all of which use a comparable Laplacian scheme for horizontal viscosity.





| $\Delta x$ (km) | $nz$ | $\Delta t$ (s) | $\nu_h$ (m$^2$s$^{-1}$) | Re$_h$ |
|---|---|---|---|---|
| 10 | 26 | 348.39 | 10.0 | 100 |
| 10 | 26 | 348.39 | 20.0 | 50 |
| 10 | 26 | 348.39 | 50.0 | 20 |
| 10 | 26 | 348.39 | 125.0 | 8 |
| 10 | 26 | 348.39 | 200.0 | 5 |
| 10 | 26 | 348.39 | 500.0 | 2 |
| 4 | 40 | 140.26 | 4.0 | 100 |
| 4 | 40 | 140.26 | 8.0 | 50 |
| 4 | 40 | 140.26 | 20.0 | 20 |
| 4 | 40 | 140.26 | 50.0 | 8 |
| 4 | 40 | 140.26 | 200.0 | 2 |

**Table 2.** Experimental setup for baroclinic eddy test case. Listed are the horizontal mesh resolution (min. triangle edge length), number of vertical levels, time step, horizontal viscosity, and the approximate grid Reynolds number.

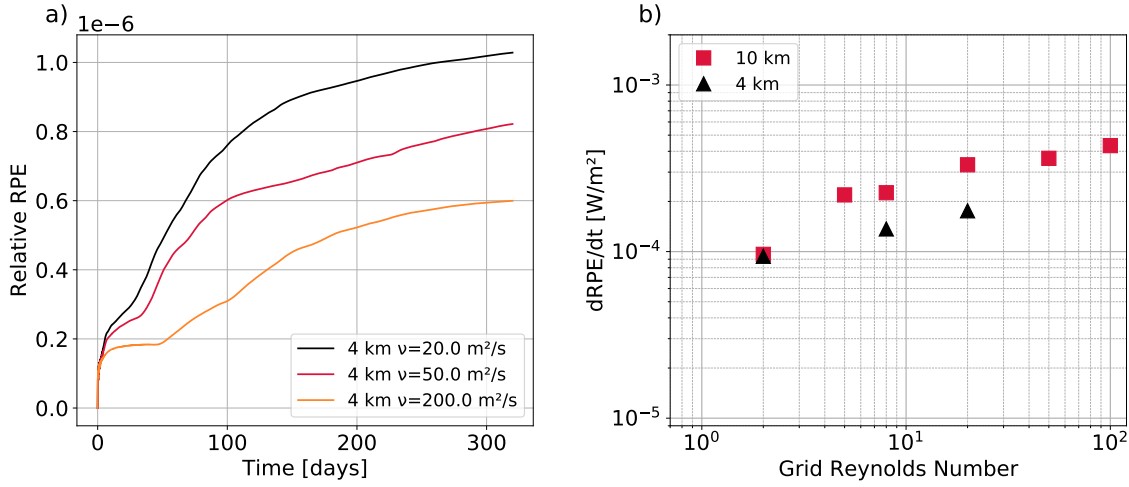

**Figure 6.** Diagnostics of the eddying channel test case. (a) Evolution of normalized RPE over time in the eddying channel test case for 4 km mesh resolution. (b) Rate of change of RPE for different grid resolutions and grid Reynolds numbers. The rate of change was evaluated by computing the average RPE change from day 2 to 320.

The evolution of the normalized RPE during the simulation is shown in Figure 6 (a) for the 4 km mesh resolution. The amount of mixing clearly depends on the grid Reynolds number, RPE being roughly twice as high for Re$_h = 20$ compared to Re$_h = 2$. The average rate of change of RPE, averaged over days 2 to 320, is shown in Figure 6 (b) for all the simulations. As expected, the rate of change increases with larger grid Reynolds number, and with a coarser mesh. These RPE metrics are in





good agreement with results in the literature: At $\mathrm{Re}_h = 20$ Thetis $d\mathrm{RPE}/dt$ values are $3.3 \times 10^{-4}$ and $1.8 \times 10^{-4}$ W m$^{-2}$, for the 10 and 4 km resolutions. The corresponding values for MITgcm, MOM, and POP (averaged over the days 3 to 319) are larger, at least $8 \times 10^{-4}$ and $3 \times 10^{-4}$ W m$^{-2}$, respectively (Petersen et al., 2015, fig. 12). Ilıcak et al. (2012) reported similar values for MITgcm and MOM. On a hexagonal mesh, MPAS-Ocean yields smaller $d\mathrm{RPE}/dt$ values, approximately $2 \times 10^{-4}$

and $7 \times 10^{-5}$ W m$^{-2}$ for the two resolutions, respectively (values averaged over days 1–320; see fig. 12 in Petersen et al., 2015). With a quad mesh, however, MPAS-Ocean values are approximately $2 \times 10^{-4}$ W m$^{-2}$ for both resolutions, therefore close to Thetis performance.

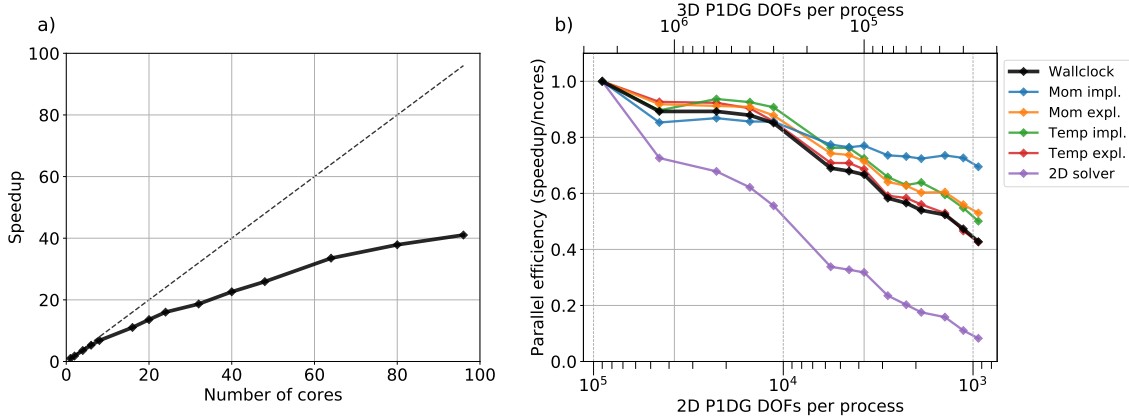

**Figure 7.** Strong parallel scaling for the baroclinic eddies test case on a 4 km mesh ($\nu_h = 20$ m$^2$s$^{-1}$): (a) Speedup in wallclock time versus number of processes; (b) Parallel efficiency versus the number of degrees of freedom (DOF) in the 3D tracer field (top axis) and the 2D $(\bar{\boldsymbol{u}}, \eta)$ mixed system (bottom axis). The black line is the wall clock time; colored lines stand for the time spent in different implicit or explicit solvers. The mesh consisted of 10 000 triangles, 40 vertical levels and 400 000 prisms.

The test cases were run on a Linux cluster with 16-core Intel Xeon E5620 processors and Mellanox Infiniband interconnect. The 320-day simulation took roughly 42 hours to run on 96 cores with the 4 km resolution mesh and 140.26 s time step. We

also carried out a strong scaling test with the 4 km mesh. In the scaling test, the simulation was run for 40 time steps, recording the total elapsed wall clock time and time spent in different parts of the solver. Figure 7 (a) shows the overall speedup up to 96 processors. The scaling efficiency drops to roughly 50% at 96 cores, when the local degree of freedom count for the tracer field is 25 000 (see black line Figure 7 b). This scaling efficiency is close to typical Firedrake performance (Rathgeber et al., 2016).

The scaling efficiency of the separate solvers is plotted with colored lines in Figure 7 (b). The implicit vertical diffu-

15 sion/viscosity solvers perform best due to the fact that the problem is purely local without any horizontal dependencies. The explicit momentum solver scales almost as well, whereas the explicit tracer solver scales poorer. The implicit 2D solver (assembly and linear solve) scales the poorest because the problem is relatively small; at 96 cores there are only around 940 degrees of freedom in the $(\bar{\boldsymbol{u}}, \eta)$ system per core. We have also experimented with explicit 2D solvers but they do not scale significantly better compared to the two-stage implicit scheme used herein.



## 5.5 DOME

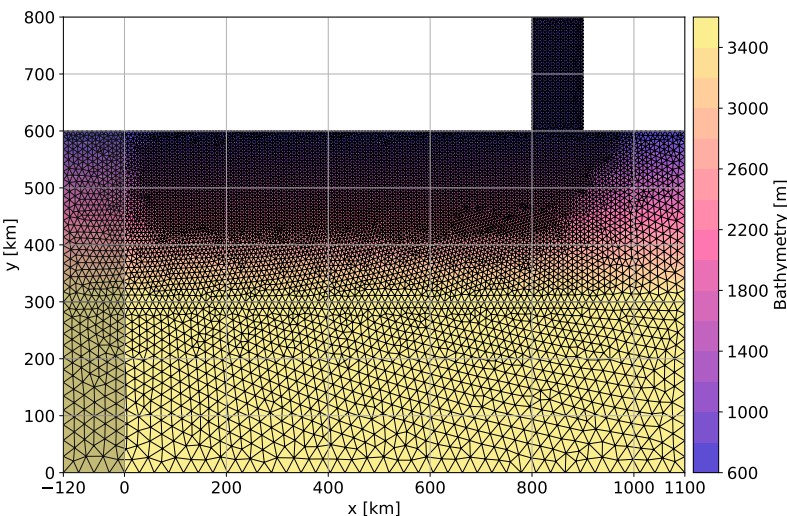

**Figure 8.** Horizontal mesh and bathymetry for the DOME test case. The domain is extended 120 km further to the west to avoid boundary effects (shaded region). Horizontal element size ranges from 6 to 22 km. There are $18.8 \times 10^3$ triangles in the horizontal mesh and 24 uniformly distributed vertical levels resulting in $450 \times 10^3$ prisms and $2.7 \times 10^6$ tracer degrees of freedom.

Next we investigate the model's ability to simulate density driven overflows with the Dynamics of Overflow Mixing and Entrainment (DOME) benchmark (Ezer and Mellor, 2004; Legg et al., 2006; Wang et al., 2008b; Burchard and Rennau, 2008). The domain is a 1100 km by 600 km large basin, whose depth varies linearly from 600 m at the northern boundary to 3600 m in the middle of the domain (see Figure 8). To avoid boundary condition issues we have extended the domain to the west by 120 km. At the northern boundary, there is a 100 km wide and 200 km long inlet. Initially, the basin is stably stratified with a linear temperature variation from 10 °C in the deepest part of the basin to 20 °C at the surface. We use the linear equation of state with $\rho_0 = 1000$ kg m$^{-3}$, $\alpha_T = 0.2$ kg m$^{-3}$ °C$^{-1}$ and $T_0 = 10$ °C resulting in a $\Delta\rho = 2.0$ kg m$^{-3}$ density difference.

At the inlet, a dense inflow (temperature 10 °C) is prescribed in the bottom layer, with the surface layer being at 20 °C. The inflow is in geostrophic balance, the thickness of the bottom layer being roughly 300 m in the eastern end of the boundary diminishing exponentially westward (Legg et al., 2006). The total inflow in the bottom layer is 5 Sv ($5 \times 10^6$ m$^3$/s), the surface layer being static. During the simulation, the fate of the inflowing waters is tracked with a passive tracer that is initially zero in the basin and unity at the inlet. Initially, the tracer field is set to the inflow conditions in the northern part of the basin ($y > 650$ km). Velocity is set to zero everywhere. The eastern and southern boundaries of the basin are closed. The western boundary is open with radiation boundary conditions, and a 100 km wide band where the temperature is relaxed to the initial condition.

The domain is discretized with an unstructured grid (Figure 8). Horizontal mesh resolution is 6 km near the northern boundary, increasing southward. 24 vertical sigma levels are used. Over the slope, the mesh resolution was designed to result in a

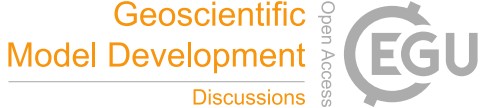



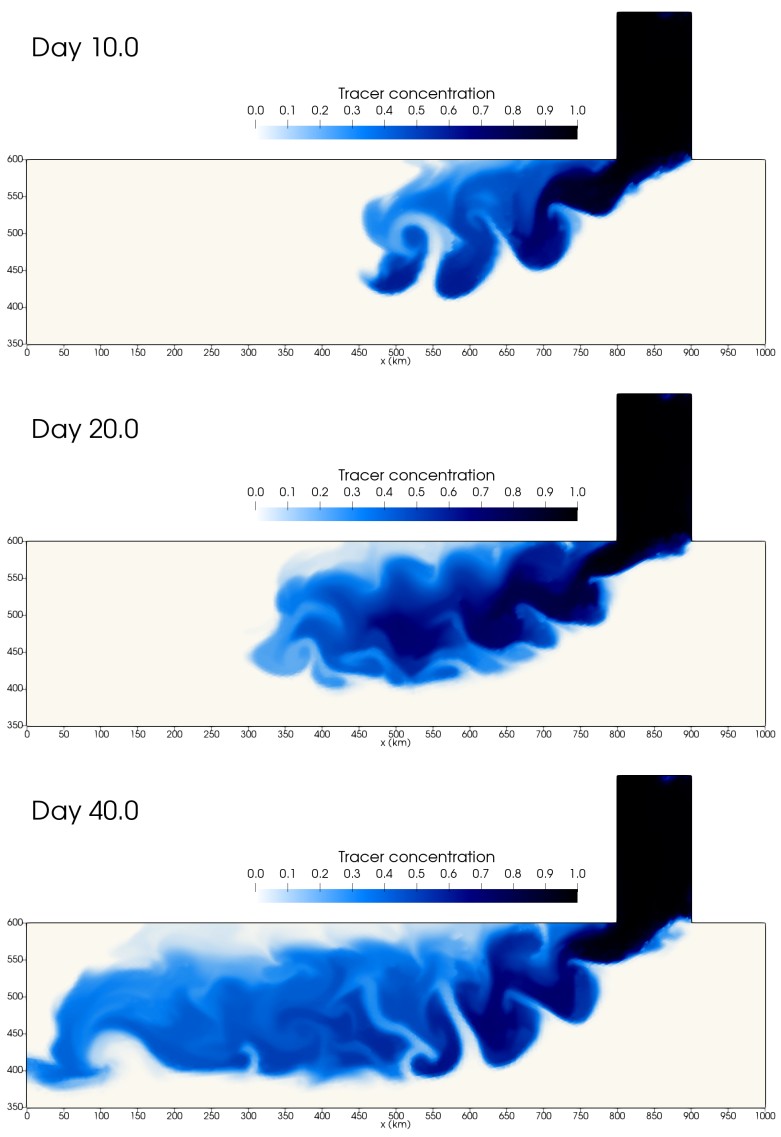

**Figure 9.** Bottom tracer concentration in the DOME test case after 10 (top), 20 (middle) and 40 (bottom) days.

hydrostatic consistency metric $r < 1.5$ (Beckmann and Haidvogel, 1993). Horizontal viscosity is set to a constant $50 \ \mathrm{m^2 s^{-1}}$, which corresponds to $\mathrm{Re}_h \approx 200$ at the inlet. Horizontal diffusivity is constant at $10 \ \mathrm{m^2 s^{-1}}$. Vertical viscosity and diffusivity are parametrized by the Pacanowski-Philander scheme as described in Section 2.4. Bottom friction is parametrized with a quadratic drag coefficient $C_d = 2 \times 10^{-3}$ (Legg et al., 2006; Wang et al., 2008b). A quadratic function space is used for the baroclinic head and internal pressure gradient as discussed in Section 3.7.





As the inflowing current reaches the basin, it turns to the west and forms a coastal plume that is approximately 150 km wide (Figure 9). The plume detaches from the lateral boundary as it flows westward and along the bottom slope. As the dense water mass meets the stratified ocean, the plume becomes unstable and starts to generate eddies and internal waves. The most vigorous eddies are found in the first 300 km after the inlet ($x = 500$–$800$ km), after which the plume is more mixed and

quiescent. Overall the plume is shallow; most of the passive tracer is concentrated within 200 m of the bottom. Qualitatively, the extent and propagation of the plume, and its eddy structure are in good agreement with the literature (e.g., Burchard and Rennau, 2008; Wang et al., 2008b). The results show that Thetis is able to represent eddying flows over sloping bathymetry, generating and maintaining strong gradients between water masses. The sharpest fronts in the simulation encompass only one or two elements.

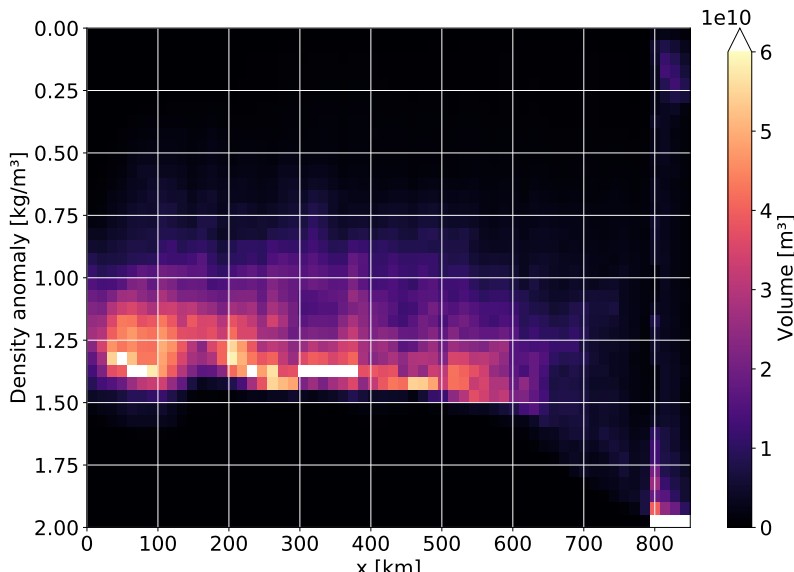

**Figure 10.** Histogram of tracer in the DOME test case versus the $x$ coordinate and density class. At the mouth of the inlet ($x = 800$ km) the inflowing waters are dense; they get entrained higher up in the density spectrum as they are being transported downstream. The data are averaged over one week after day 40.

Figure 10 shows the distribution of the inflowing tracer concentration as a function of water density and the $x$-axis. The inflowing waters are initially very dense but get mixed to lower density as the plume advances along the coast. The histogram shows that the plume volume is low in the first 150 km after the inlet ($x = 650$–$800$ km) where the plume accelerates. After $x = 650$ km the plume slows down and starts to accumulate in volume. The density of the main plume occupies ranges from $0.8$ to $1.5$ kg m$^{-3}$, the peak being around $1.28$ kg m$^{-3}$. The rate of entrainment can be used as a metric for mixing. Results

herein are similar to those presented in literature: Wang et al. (2008b) present a mean density anomaly of $1.5$ kg m$^{-3}$ for their terrain following FESOM model configuration.

The 47-day simulation took roughly 42 hours to run on 90 cores with a 39.65 s time step on the same Linux cluster.



## 6 Conclusions

This paper describes a DG implementation of an eddy-permitting, unstructured grid coastal ocean model. The solver is second-order accurate in space and time. We have demonstrated that the formulation is fully conservative and preserves monotonicity. The test cases indicate that the model is capable of reproducing the expected physical behavior, including baroclinic eddies. Moreover, numerical mixing is well-controlled and comparable to other established structured grid models, such as MITgcm and ROMS, and the large-scale finite volume model MPAS-Ocean. Finding an accurate formulation is important as commonly-used unstructured grid models tend to be overly diffusive, preventing accurate modeling of certain coastal domains (e.g., Kärnä et al., 2015). The formulation presented herein thus contributes to the development of more accurate next-generation coastal ocean models.

Future work will include solving the equations on a sphere, DG implementation of a biharmonic viscosity operator, two-equation turbulence closure models, wetting-drying treatment, development of an adjoint solver, as well as improving the computational efficiency and parallel scaling of the solver.

## 7 Code availability

The packages used to perform the experiments have been archived using Zenodo: Thetis (zenodo/thetis), Firedrake (zenodo/firedrake), PETSc (zenodo/petsc), petsc4py (zenodo/petsc4py), COFFEE (zenodo/coffee), FIAT (zenodo/fiat), FInAT (zenodo/finat), PyOP2 (zenodo/pyop2), TSFC (zenodo/tsfc), and UFL (zenodo/ufl). The source code repositories as well as the archived versions are publicly available.

## 8 Data availability

No external data were used in this manuscript.

## Appendix A: Source terms for the baroclinic MMS test

Using the analytical velocity and temperature fields we can derive the steady state solution for the remaining fields

$$\eta_a = 0, \tag{A1}$$

$$\bar{u}_a = \frac{1}{6}\sin(3)\sin\left(\frac{2\pi}{L_x}x\right), \tag{A2}$$

$$\bar{v}_a = \frac{1}{3}\sin\left(\frac{z}{2h}\right)\cos\left(\frac{\pi y}{L_y}\right), \tag{A3}$$

$$u'_a = u_a - \bar{u}_a, \tag{A4}$$

$$v'_a = v_a - \bar{v}_a, \tag{A5}$$





$$w_a = \frac{\pi h}{3 L_x L_y} \left( 2 L_x \left( -\cos\left(\frac{z}{2h}\right) + \cos\left(\frac{1}{2}\right) \right) \sin\left(\frac{\pi y}{L_y}\right) - L_y \left( \sin\left(\frac{3z}{h}\right) + \sin(3) \right) \cos\left(\frac{2\pi}{L_x} x\right) \right), \tag{A6}$$

$$r_a = \frac{\alpha_T}{\rho_0} \left( T_0 z - 15 h \sin\left(\frac{z}{h}\right) \sin\left(\frac{\pi x}{L_x}\right) \sin\left(\frac{\pi y}{L_y}\right) - 15 z \right). \tag{A7}$$

Now we can evaluate the different terms that appear in the momentum and tracer equations:

$$f(\boldsymbol{e}_z \wedge \bar{\boldsymbol{u}})_x = \frac{2 f_0}{3} \left( -\cos\left(\frac{1}{2}\right) + 1 \right) \cos\left(\frac{\pi y}{L_y}\right), \tag{A8}$$

$$f(\boldsymbol{e}_z \wedge \bar{\boldsymbol{u}})_y = \frac{f_0}{6} \sin(3) \sin\left(\frac{2\pi}{L_x} x\right), \tag{A9}$$

$$\boldsymbol{\nabla}_h \cdot (H \bar{\boldsymbol{u}}) = \frac{\pi h}{3 L_x L_y} \left( 2 L_x \left( -\cos\left(\frac{1}{2}\right) + 1 \right) \sin\left(\frac{\pi y}{L_y}\right) + L_y \sin(3) \cos\left(\frac{2\pi}{L_x} x\right) \right), \tag{A10}$$

$$(\boldsymbol{F}_{\text{pg}})_x = \frac{15 \pi \alpha_T h}{L_x \rho_0} g \sin\left(\frac{z}{h}\right) \sin\left(\frac{\pi y}{L_y}\right) \cos\left(\frac{\pi x}{L_x}\right), \tag{A11}$$

$$(\boldsymbol{F}_{\text{pg}})_y = \frac{15 \pi \alpha_T h}{L_y \rho_0} g \sin\left(\frac{z}{h}\right) \sin\left(\frac{\pi x}{L_x}\right) \cos\left(\frac{\pi y}{L_y}\right), \tag{A12}$$

$$(\boldsymbol{\nabla}_h \cdot (\boldsymbol{u} \boldsymbol{u}))_x = \frac{\pi}{2 L_x} \sin\left(\frac{2\pi}{L_x} x\right) \cos^2\left(\frac{3z}{h}\right) \cos\left(\frac{2\pi}{L_x} x\right), \tag{A13}$$

$$(\boldsymbol{\nabla}_h \cdot (\boldsymbol{u} \boldsymbol{u}))_y = -\frac{\pi}{9 L_y} \sin^2\left(\frac{z}{2h}\right) \sin\left(\frac{\pi y}{L_y}\right) \cos\left(\frac{\pi y}{L_y}\right), \tag{A14}$$

$$\frac{\partial (wu)}{\partial z} = \frac{\pi \sin\left(\frac{3z}{h}\right)}{2 L_x L_y} \left( 2 L_x \left( \cos\left(\frac{z}{2h}\right) - \cos\left(\frac{1}{2}\right) \right) \sin\left(\frac{\pi y}{L_y}\right) + L_y \left( \sin\left(\frac{3z}{h}\right) + \sin(3) \right) \cos\left(\frac{2\pi}{L_x} x\right) \right) \sin\left(\frac{2\pi}{L_x} x\right), \tag{A15}$$

$$\frac{\partial (wv)}{\partial z} = -\frac{\pi \cos\left(\frac{z}{2h}\right)}{18 L_x L_y} \left( 2 L_x \left( \cos\left(\frac{z}{2h}\right) - \cos\left(\frac{1}{2}\right) \right) \sin\left(\frac{\pi y}{L_y}\right) + L_y \left( \sin\left(\frac{3z}{h}\right) + \sin(3) \right) \cos\left(\frac{2\pi}{L_x} x\right) \right) \cos\left(\frac{\pi y}{L_y}\right), \tag{A16}$$

$$f(\boldsymbol{e}_z \wedge \boldsymbol{u}')_x = \frac{f_0}{3} \left( -\sin\left(\frac{z}{2h}\right) - 2 + 2\cos\left(\frac{1}{2}\right) \right) \cos\left(\frac{\pi y}{L_y}\right), \tag{A17}$$

$$f(\boldsymbol{e}_z \wedge \boldsymbol{u}')_y = \frac{f_0}{6} \left( 3\cos\left(\frac{3z}{h}\right) - \sin(3) \right) \sin\left(\frac{2\pi}{L_x} x\right), \tag{A18}$$

$$\boldsymbol{\nabla}_h \cdot (\boldsymbol{u} T) = \frac{5 \pi}{L_x L_y} \left( L_x \sin\left(\frac{z}{2h}\right) \cos^2\left(\frac{\pi y}{L_y}\right) + 3 L_y \sin\left(\frac{\pi y}{L_y}\right) \cos\left(\frac{3z}{h}\right) \cos^2\left(\frac{\pi x}{L_x}\right) \right) \sin\left(\frac{\pi x}{L_x}\right) \cos\left(\frac{z}{h}\right), \tag{A19}$$

$$\frac{\partial (wT)}{\partial z} = \frac{5 \pi}{L_x L_y} \left[ 2 L_x \left( \cos\left(\frac{z}{2h}\right) - \cos\left(\frac{1}{2}\right) \right) \sin\left(\frac{\pi y}{L_y}\right) + \right.$$
$$\left. L_y \left( \sin\left(\frac{3z}{h}\right) + \sin(3) \right) \cos\left(\frac{2\pi}{L_x} x\right) \right] \sin\left(\frac{z}{h}\right) \sin\left(\frac{\pi x}{L_x}\right) \sin\left(\frac{\pi y}{L_y}\right). \tag{A20}$$



These terms are added as source terms to the right hand side of the equations (9), (10), (11), and (6). In the weak form this corresponds to multiplying the analytical function by the test function and integrating over the domain. The solutions were derived using the SymPy symbolic mathematics Python library (Meurer et al., 2017).

*Author contributions.* Tuomas Kärnä designed and implemented most of the solver and carried out the numerical simulations. Stephan
5  Kramer and Lawrence Mitchell contributed to the design and implementation of the model. António Baptista, David Ham and Matthew Piggott supervised the work and guided the implementation of the model and the manuscript.

*Acknowledgements.* The National Science Foundation partially supported this research through cooperative agreement OCE-0424602. The National Oceanic and Atmospheric Administration (NA11NOS0120036 and AB-133F-12-SE-2046), Bonneville Power Administration (00062251) and Corps of Engineers (W9127N-12-2-007 and G13PX01212) provided partial motivation and additional support. This
10  work was supported by the UK's Engineering and Physical Science Research Council [grant numbers EP/M011054/1, EP/L000407/1]; and the Natural Environment Research Council [grant number NE/K008951/1]. This work used the Extreme Science and Engineering Discovery Environment (XSEDE), which is supported by National Science Foundation grant number ACI-1053575. The authors acknowledge the Texas Advanced Computing Center (TACC) at The University of Texas at Austin for providing HPC resources that have contributed to the research results reported within this paper.



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
