# Peer review of "Thetis coastal ocean model: discontinuous Galerkin discretization for the three-dimensional hydrostatic equations"

_Geoscientific Model Development, 2017_

## Short Comment (SC1) · 20 Mar 2018

Dear authors,

in my role as Executive editor of GMD, I would like to bring to your attention our Editorial version 1.1:

http://www.geosci-model-dev.net/8/3487/2015/gmd-8-3487-2015.html

This highlights some requirements of papers published in GMD, which is also available on the GMD website in the 'Manuscript Types' section:

http://www.geoscientific-model-development.net/submission/manuscript_types.html

In particular, please note that for your paper, the following requirement has not been met in the Discussions paper:

- "The main paper must give the model name and version number (or other unique identifier) in the title."

Please add a version number for Thetis to the title upon your revised submission to GMD.

Yours,

Astrid Kerkweg

---

## Author Comment (AC1) · 29 Mar 2018

Dear Executive Editor,

Thank you for your comment. We recognize that GMD requires both the model name and version number in the manuscript title. We have already included the model name. However, Thetis does not have version numbers as a matter of policy. We rely on continuous release paradigm which implies that the only supported version is the master version available online. For the purpose of scientific reproducibility, we do archive particular versions of the source code, identified by a DOI. (This has been done in the present manuscript as well, see Section 7). Therefore the only "version number" that

we could include in the title is the DOI, which is not a very convenient solution. We, therefore, request that the title could be accepted as it is.

As a precedent, we note that GMD has previously accepted a similar argument for a Firedrake paper: https://www.geosci-model-dev.net/9/3803/2016/gmd-9-3803-2016-discussion.html

On behalf of the authors, Tuomas Karna

---

## Editor Comment (EC1) · J. D. Annan (Editor) · 5 Apr 2018

The absence of version number is noted, and it seems to me that adding the DOI in the title would be a simple and straightforward way of solving this problem, eg "Thetis coastal ocean model (doi): discontinuous...". I don't see any problem at all with having a doi in the title if it is the only way to refer to the code. GMD imposes no requirements on authors to support their code.

Additionally, the references to the source code in the code availability section seem a little obscure. It did not occur to me that "zenodo/fiat" would be a reference entry and searching for fiat on zenodo was insufficient. Again, including the doi in the text would

work well here, or else a sentence to explain where the doi could be found.

---

## Author Comment (AC2) · 18 Apr 2018

Dear James D. Annan,

First of all, I need to apologize for my poor wording in my reply: Adding a DOI in the manuscript title is not really an option for us. We have explicitly chosen a continuous release paradigm where we do not assign version numbers at all. Adding a version number in the manuscript title (in any format, DOI, or other) would thus mislead the reader to think that there exists a fixed release version that can be referred to. This is not the case.

[Figure]

About the references to the source code, we will clarify how the code can be obtained via the Zenodo service in Section 7 of the revised manuscript.

Yours sincerely,

Tuomas Kärnä
* * *

---

## Editor Comment (EC2) · J. D. Annan (Editor) · 26 Apr 2018

Thank you for the reply. Making the code availability section clearer will be appreciated.

I have considered the issue of the title/version number carefully and discussed it with some colleagues. While I consider that putting the DOI in the title would be the best way to fulfil the spirit of the GMD guidelines in this case (I find the argument that this is inconvenient to be unconvincing), I have also always been sympathetic to the view that "rules are for the obedience of fools, and the guidance of wise men". In this case, with no specific code release, the only purpose of the DOI or other code identifier would be for the replication of this paper's results and there is essentially zero chance that

anyone else would want to use the exact same code for different purposes. No-one else is likely to recognise, or search for, the DOI, in the way that they might in the case of a released "Model x.y". Thus, I agree with the authors that the requirement can be waived in this case. I would like to point out that this decision is entirely my own and other editors may think differently in similar circumstances in the future.

———————————————

---

## Referee Comment (RC1) · Anonymous Referee #1 · 31 May 2018

The manuscript is well written, and I would recommend it after the points below are addressed.

1. I appreciate the idea of reducing dissipation. However, I haven't found any special measures specifically devoted to that. The dissipation introduced through the Lax–Friedrichs flux is applied everywhere, which is approximately equivalent to saying that the Reynolds or Peclet numbers on the grid scale are about one. How dissipation related to this flux compares to the explicit dissipation introduced in the code? I think it could be a good message to community if the authors will manage to demonstrate that dissipation due to numerical fluxes in low-order DG code is not too strong. Common

wisdom in ocean modeling is that the horizontal viscosity is selected as Vh, where h is the grid scale and V about 1 cm/s. Can the authors propose an estimate of effective viscosity in their code?

2. Significant part of dissipation in coastal codes can be traced back to friction added to barotropic equation to stabilize the barotropic flow in wetting-drying regimes. I do not see this in the present model, and would recommend to comment on that in the manuscript. In two-stage procedure: I do not see that the first solve for the elevation is implicit (Eq. 46). Please clarify this place. Time step limitations: I find the discussion to be a bit superficial, the CFL limitations in 2D are not the same as in 1D, and it is net limitation of horizontal and vertical advection that matters.

3. Scalability: From Fig. 7 I can conclude that scaling efficiency is on the level of 50% already for 50 cores. The mesh used contains 5k vertices, giving 100 vertices per core. This level is very good, however it is achieved even with some finite-volume codes such as MPAS atmosphere (I do not have information on MPAS-ocean). The point is that with DG one expects more floating point operations on the local level, i. e. better scalability, which is not the case. Bad scalability of 2D solver is noteworthy and is against expectations. Is it PETSc on its own, or the assembly operations? How preconditioning is organized? Some critical analysis is needed. In recent finite-volume ocean linear scaling is maintained 300-400 vertices per core, and here I see that the DG case it is not any better! Of course it depends on interconnect, but I do not see the message I expected: that DG codes scale better than FV ones.

4. Finally, the performance. For me the numbers are really disappointing. First, I would like to see how it compares to previous efforts (SLIM, UTBEST or like). Is there any progress in computational efficiency of DG codes? Second, please compare the throughput of Thetis to the throughput of other unstructured-mesh codes (MPAS, FV-COM, SHCISM, FESOM). There are some published data. My very crude estimates give a factor from 20 to 100. I am not willing to use this as an argument against; on the contrary, I would like to propose to critically analyse the performance and try to answer

why DG codes are that slow and what are the promises. In most cases it is the writing into memory or taking data from memory that limits the performance. Is it the mere enhanced size of DoF in DG codes? I think it would be a very valuable addition. Then, there is a question on effective resolution. Does the much larger number of DoFs in DG leads to better effective resolution than say MPAS approach? I do realize that the last question deserves a separate study and is not in the scope of GMD, but once again, I am missing the perspective. On the practical level of using the codes a user would be interested in throughput. It can be reached (i) directly or (ii) through better scalability or (iii) through better effective resolution. Is there any hope that a combination of these would make the DG codes same practical as their FV counterparts?

---

## Referee Comment (RC2) · J. D. Annan (Referee) · 27 Aug 2018

I'm reluctantly forced into the position of acting as reviewer due to a lack of other options. In fact our of the numerous requests made, two other referees did promised to provide reviews (in addition to the one already published) but they have subsequently stopped responding to email.

However, the paper is basically well-written and we already have one informed and careful review so in this case I'm comfortable proceeding on the basis of this and my own views.

[Figure]

The manuscript presents an interesting approach to unstructured grids (including a free surface but not moving in the horizontal or otherwise adaptive). Are there any plans to extend to adaptive grids, other than what might be implied by wetting and drying?

The paper presents several standard tests all of which appear to produce acceptable results, and conforms to the GMD standards (noting the earlier discussion concerning title/code). Therefore I would be happy to recommend publication after the minor revisions required by reviewer 1.
* * *

---

## Author Comment (AC4) · 4 Sep 2018

We would like to thank the reviewer for their positive comments.

*The manuscript presents an interesting approach to unstructured grids (including a free surface but not moving in the horizontal or otherwise adaptive). Are there any plans to extend to adaptive grids, other than what might be implied by wetting and drying?*

Mesh adaptivity is currently being implemented in the Firedrake modeling framework. Once completed, we are indeed planning on introducing horizontal mesh adaptivity in

Thetis as well.

---

## Author Response (AR1)

September 4, 2018

**1 Anonymous Referee #1**

The authors would like to sincerely thank the referee for the careful review and constructive comments that have helped to improve the manuscript.

*The manuscript is well written, and I would recommend it after the points below are addressed.*

*1. I appreciate the idea of reducing dissipation. However, I haven't found any special measures specifically devoted to that. The dissipation introduced through the Lax–Friedrichs flux is applied everywhere, which is approximately equivalent to saying that the Reynolds or Peclet numbers on the grid scale are about one. How dissipation related to this flux compares to the explicit dissipation introduced in the code? I think it could be a good message to community if the authors will manage to demonstrate that dissipation due to numerical fluxes in low-order DG code is not too strong. Common wisdom in ocean modeling is that the horizontal viscosity is selected as Vh, where h is the grid scale and V about 1 cm/s. Can the authors propose an estimate of effective viscosity in their code?*

We are indeed using the Lax-Friedrichs (LF) flux in the model. In contrast to the first version of the manuscript we are now using the LF flux in the momentum equation, but have omitted it from the tracer equation. This reduces RPE in some test cases (e.g. lock exchange) but does not significantly change the overall performance of the model. All the numerical results have been re-generated and the manuscript has been updated accordingly.

[Figure]

Figure 1: Lock exchange test with different values of viscosity and either excluding (solid lines) or including (dashed lines) the Lax–Friedrichs flux.

To address the influence of the LF flux on numerical mixing, we ran the lock exchange test varying the viscosity, and either including or excluding the LF flux (see Fig 1). The RPE values obtained with zero

viscosity and the LF flux are close to RPE obtained with $\nu = 3.125 \ \mathrm{m}^2/s$ and no LF flux. The viscosity value corresponds to $Re = 80$. In addition, it is evident that for $Re < 10$ ($\nu > 50$) the LF flux has practically no effect on the RPE. Thus (in this particular test case) the LF flux introduces mixing that's roughly equivalent to $3 \ \mathrm{m}^2/s$ viscosity, or $Re = 80$. More generally one can argue that the LF flux has negligible impact on numerical mixing if $Re < 10$.

*2. Significant part of dissipation in coastal codes can be traced back to friction added to barotropic equation to stabilize the barotropic flow in wetting-drying regimes. I do not see this in the present model, and would recommend to comment on that in the manuscript. In two-stage procedure: I do not see that the first solve for the elevation is implicit (Eq. 46). Please clarify this place. Time step limitations: I find the discussion to be a bit superficial, the CFL limitations in 2D are not the same as in 1D, and it is net limitation of horizontal and vertical advection that matters.*

It is true that wetting and drying schemes may introduce a significant amount of dissipation. As we are not considering wetting and drying in this paper we have not addressed this issue directly. However, we do mention wetting-drying induced dissipation in the introduction of the revised manuscript.

In terms of equation (46) (or 44), the first solve is indeed explicit. We note that the second stage, equation (45), is just a Crank-Nicolson update, and hence the first stage result is not used in the final stage. We are only writing the system (44-45) as a two-stage system in order to combine it with 2nd order SSP scheme used in the 3D mode.

In terms of the time step limitations, we have revised Section 4.3. The 2D geometry is taken into account by the appropriately chosen mesh size metric $L_h$ and the scaling factor $\sigma$ which depends on the shape of the element, polynomial degree, and the accuracy of the RK time integration scheme. We agree that in general the time step is limited by the net horizontal and vertical advection. However, as $w$ remains relatively small in the presented test cases there is no need to formulate the CFL constraint for the 3D $(\boldsymbol{u}, w)$ velocity vector.

*3. Scalability: From Fig. 7 I can conclude that scaling efficiency is on the level of 50% already for 50 cores. The mesh used contains 5k vertices, giving 100 vertices per core. This level is very good, however it is achieved even with some finite-volume codes such as MPAS atmosphere (I do not have information on MPAS-ocean). The point is that with DG one expects more floating point operations on the local level, i. e. better scalability, which is not the case. Bad scalability of 2D solver is noteworthy and is against expectations. Is it PETSc on its own, or the assembly operations? How preconditioning is organized? Some critical analysis is needed. In recent finite-volume ocean linear scaling is maintained 300-400 vertices per core, and here I see that the DG case it is not any better! Of course it depends on interconnect, but I do not see the message I expected: that DG codes scale better than FV ones.*

DG methods do provide better strong scaling compared to finite-volume (FV) formulation, but usually that is only expected for high-order DG. For first order DG one would expect the scaling performance to be close to FV methods.

It should be noted that the strong scaling results are affected by Firedrake overhead, related to Python and the parallel scheduler (PyOP2) overheads. We have not yet fully addressed these issues and believe that the scaling can be improved significantly in the future. The main purpose of the paper is to present the discretization; we provide the performance metrics only for the sake of completeness.

The poorer 2D solver performance is due to the fact that the 2D problem is (significantly) smaller (see the bottom horizontal axis in Figure 7 of the revised manuscript). In fact, in terms of the DOFs per core, the 2D solver scales a bit better than most of the 3D solvers. The cost is mostly in the solver and preconditioner. We use PETSc GMRES solver with simple multiplicative field-split preconditioner. In the future the performance of the 2D solver could be improved by using hybridized DG methods for instance, but that is out of the scope of the present paper.

*4. Finally, the performance. For me the numbers are really disappointing. First, I would like to see how it compares to previous efforts (SLIM, UTBEST or like). Is there any progress in computational efficiency of DG codes? Second, please compare the throughput of Thetis to the throughput of other unstructured-mesh codes (MPAS, FV-COM, SHCISM, FESOM). There are some published data. My very crude estimates give a factor from 20 to 100. I am not willing to use this as an argument against; on the contrary, I would like to propose to critically analyse the performance and try to answer why DG*

*codes are that slow and what are the promises. In most cases it is the writing into memory or taking data from memory that limits the performance. Is it the mere enhanced size of DoF in DG codes? I think it would be a very valuable addition. Then, there is a question on effective resolution. Does the much larger number of DoFs in DG leads to better effective resolution than say MPAS approach? I do realize that the last question deserves a separate study and is not in the scope of GMD, but once again, I am missing the perspective. On the practical level of using the codes a user would be interested in throughput. It can be reached (i) directly or (ii) through better scalability or (iii) through better effective resolution. Is there any hope that a combination of these would make the DG codes same practical as their FV counterparts?*

We have added a comparison against SLIM 3D in the appendix of the revised manuscript. Comparison against SLIM 3D is straightforward as both use a similar DG formulation. Using the same mesh and time step, Thetis is 2 to 4x faster than SLIM 3D, although SLIM 3D is written in C/C++ and Thetis uses Python at runtime. Better Thetis performance is likely related to Firedrake's better memory layout of 3D fields, and efficient code generation. We believe that the performance can be further improved in the future both due to improvements in Firedrake and Thetis optimizations.

We also note that in the test cases the time step is set below the CFL limit, and thus the reported the wall clock times are not directly representative of the computational efficiency.

We agree that comparing the performance and accuracy of Thetis against other established unstructured grid models is absolutely necessary. This task is, however, not trivial: the experiments and accuracy metrics should be designed carefully. Based on the timings it is evident that Thetis is slower than other models, e.g. SCHISM. However, our preliminary tests do suggest that it is also more accurate (not shown). Thus, as the Reviewer suggests, we agree that having a robust metric for the effective resolution is crucial for carrying out such a comparison. As such, model inter-comparison is too big of a task to be included in the present paper but we aspire to address it in the future.

**2 Referee #2**

*I'm reluctantly forced into the position of acting as reviewer due to a lack of other options. In fact our of the numerous requests made, two other referees did promised to provide reviews (in addition to the one already published) but they have subsequently stopped responding to email.*

*However, the paper is basically well-written and we already have one informed and careful review so in this case I'm comfortable proceeding on the basis of this and my own views.*

*The manuscript presents an interesting approach to unstructured grids (including a free surface but not moving in the horizontal or otherwise adaptive). Are there any plans to extend to adaptive grids, other than what might be implied by wetting and drying?*

Mesh adaptivity is currently being implemented in the Firedrake modeling framework. Once completed, we are indeed planning on introducing horizontal mesh adaptivity in Thetis as well.

*The paper presents several standard tests all of which appear to produce acceptable results, and conforms to the GMD standards (noting the earlier discussion concern- ing title/code). Therefore I would be happy to recommend publication after the minor revisions required by reviewer 1.*

**List of Changes**

September 4, 2018

List of changes since the first submission:

- Lax–Friedrichs flux is only applied in the momentum equation, but not in the tracer equation. All numerical tests have been re-done.

- Time step discussion has been revised in Section 4.3.

- Lax–Friedrichs flux induced mixing is disussed in Section 5.3.

- A CPU cost comparison against SLIM 3D model is added in Appendix B with discussion in Section 5.4.

All major changes in the manuscript are indicated with red typeface.

[revised manuscript text omitted]